# What Do Neural Networks Learn When Trained With Random Labels?

**Hartmut Maennel**[*]
hartmutm@google.com

**Ibrahim Alabdulmohsin**[*]
ibomohsin@google.com

**Ilya Tolstikhin**
tolstikhin@google.com

**Robert J. N. Baldock**[†]
rbaldock@google.com

**Olivier Bousquet**
obousquet@google.com

**Sylvain Gelly**
sylvaingelly@google.com

**Daniel Keysers**
keysers@google.com

Google Research, Brain Team
Zürich, Switzerland

## Abstract

We study deep neural networks (DNNs) trained on natural image data with entirely random labels. Despite its popularity in the literature, where it is often used to study memorization, generalization, and other phenomena, little is known about what DNNs learn in this setting. In this paper, we show analytically for convolutional and fully connected networks that an alignment between the principal components of network parameters and data takes place when training with random labels. We study this alignment effect by investigating neural networks pre-trained on randomly labelled image data and subsequently fine-tuned on disjoint datasets with random or real labels. We show how this alignment produces a *positive* transfer: networks pre-trained with random labels train faster downstream compared to training from scratch even after accounting for simple effects, such as weight scaling. We analyze how competing effects, such as specialization at later layers, may hide the positive transfer. These effects are studied in several network architectures, including VGG16 and ResNet18, on CIFAR10 and ImageNet.

## 1 Introduction

Over-parameterization helps deep neural networks (DNNs) to generalize better in real-life applications [8, 24, 30, 54], despite providing them with the capacity to fit almost any set of random labels [55]. This phenomenon has spawned a growing body of work that aims at identifying fundamental differences between real and random labels, such as in training time [4, 19, 20, 56], sharpness of the minima [28, 40], dimensionality of layer embeddings [3, 11, 35], and sensitivity [4, 41], among other complexity measures [6, 7, 39, 40]. While it is obvious that over-parameterization helps DNNs to interpolate any set of random labels, it is not immediately clear what DNNs *learn* when trained in this setting. The objective of this study is to provide a partial answer to this question.

There are at least two reasons why answering this question is of value. First, in order to understand how DNNs work, it is imperative to observe how they behave under "extreme" conditions, such as

---

[*]Equal contribution.

[†]Work completed during the Google AI Residency Program.

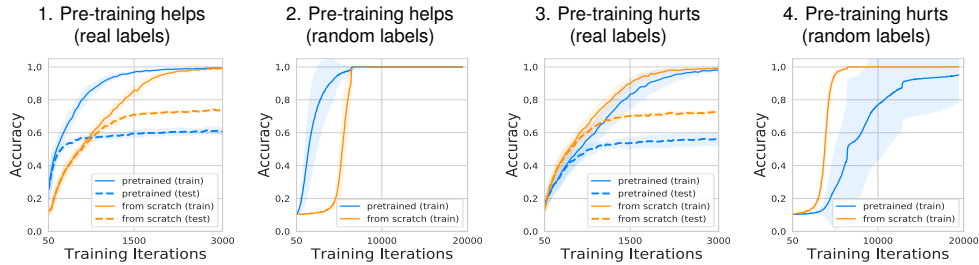

Figure 1: Pre-training on random labels may exhibit both positive (1 & 2) and negative (3 & 4) effects on the downstream fine-tuning depending on the setup. VGG16 models are pre-trained on CIFAR10 examples with random labels and subsequently fine-tuned on the fresh CIFAR10 examples with either real labels (1 & 3) or 10 random labels (2 & 4) using different hyperparameters.

when trained with labels that are entirely random. Since the pioneering work of [55], several works have looked into the case of random labels. What distinguishes our work from others is that previous works aimed to demonstrate differences between real and random labels, highlighting the *negative* side of training on random labels. By contrast, this work provides insights into what properties of the data distribution DNNs learn when trained on random labels.

Second, observing DNNs trained on random labels can explain phenomena that have been previously noted, but were poorly understood. In particular, by studying what is learned on random labels, we offer new insights into: (1) why DNNs exhibit critical stages [1, 17], (2) how earlier layers in DNNs generalize while later layers specialize [3, 4, 10, 53], (3) why the filters learned by DNNs in the first layer seem to encode some useful structure when trained on random labels [4], and (4) why pre-training on random labels can accelerate training in downstream tasks [42]. We show that even when controlling for simple explanations like weight scaling (which was not always accounted for previously), such curious observations continue to hold.

The main contributions of this work are:

- We investigate DNNs trained with random labels and fine-tuned on disjoint image data with real or random labels, demonstrating unexpected positive and negative effects.

- We provide explanations of the observed effects. We show analytically for convolutional and fully connected networks that an alignment between the principal components of the network parameters and the data takes place. We demonstrate experimentally how this effect explains why pre-training on random labels helps. We also show why, under certain conditions, pre-training on random labels can hurt the downstream task due to specialization at the later layers.

- We conduct experiments verifying that these effects are present in several network architectures, including VGG16 [46] and ResNet18-v2 [22], on CIFAR10 [31] and ImageNet ILSVRC-2012 [14], across a range of hyper-parameters, such as the learning rate, initialization, number of training iterations, width and depth.

In this work, we do not use data augmentation as it provides a (weak) supervisory signal. Moreover, we use the terms "positive" and "negative" to describe the impact of what is learned with random labels on the downstream training, such as faster/slower training. The networks reported throughout the paper are taken from a big set of experiments that we conducted using popular network architectures, datasets, and wide hyperparameter ranges. Experimental details are provided in Appendix A and B. We use boldface for random variables, small letters for their values, and capital letters for matrices.

## 1.1 Motivating example

Figure 1 shows learning curves of the VGG16 architecture [46] pre-trained on 20k CIFAR10 examples [31] with random labels (upstream) and fine-tuned on a disjoint subset of 25k CIFAR10 examples with either random or real labels (downstream). We observe that in this setup, pre-training a neural network on images with random labels accelerates training on a second set of images, both for real and random labels (positive effect). However, in the *same setting* but with a different initialization scale and number of random classes upstream, a negative effect can be observed downstream: training

becomes slower. We also observe a lower final test accuracy for real labels in both cases, which we are not explicitly investigating in this paper (and which has been observed before, e.g. in [17]).

The fact that pre-training on random labels can accelerate training downstream has been observed previously, e.g. in [42]. However, there is a "simple" property that can explain improvements in the downstream task: Because the cross-entropy loss is scale-sensitive, training the network tends to increase the scale of the weights [40], which can increase the effective learning rate of the downstream task (see the gray curve in Figure 5). To eliminate this effect, in all experiments we re-scale the weights of the network after pre-training to match their $\ell_2$ norms at initialization. We show that even after this correction, pre-training on random labels positively affects the downstream task. This holds for both VGG16 and ResNet18 trained on CIFAR10 and ImageNet (see Appendix B).

We show experimentally that some of the positive transfer is due to the second-order statistics of the network parameters. We prove that when trained on random labels, the principal components of weights at the first layer are aligned with the principal components of data. Interestingly, this *alignment effect* implies that the model parameters learned at the first layer can be summarized by a one-dimensional mapping between the eigenvalues of the data and the eigenvalues of the network parameters. We study these mappings empirically and raise some new open questions. We also analyze how, under certain conditions, a competing effect of specialization at the later layers may hide the positive transfer of pre-training on random labels, which we show to be responsible for the negative effect demonstrated in Figure 1.

To the best of our knowledge, the alignment effect has not been established in the literature before. This paper proves the existence of this effect and studies its implications. Note that while these effects are established for training on random labels, we also observe them empirically for real labels.

## 1.2 Related work

A large body of work in the literature has developed techniques for mitigating the impact of *partial* label noise, such as [56, 25, 38, 27, 32, 33, 48]. Our work is distinct from this line of literature because we focus on the case of purely random labels.

The fact that positive and negative learning takes place is related to the common observation that earlier layers in DNNs learn general-purpose representations whereas later layers specialize [3, 4, 10, 53]. For random labels, it has been noted that memorization happens at the later layers, as observed by measuring the classification accuracy using activations as features [10] or by estimating the intrinsic dimensionality of the activations [3]. We show that specialization at the later layers has a negative effect because it exacerbates the inactive ReLU phenomenon. Inactive ReLUs have been studied in previous works, which suggest that this effect could be mitigated by either increasing the width or decreasing the depth [34], using skip connections [16], or using other activation functions, such as the leaky ReLU [36, 21] or the exponential learning unit (ELU) [9].

For transfer learning, it has been observed that pre-training on random labels can accelerate training on real labels in the downstream task [42]. However, prior works have not accounted for simple effects, such as the change in first-order weight statistics (scaling), which increases when using the scale-sensitive cross-entropy loss [40]. Changing the norm of the weights alters the effective learning rate. [43] investigated transfer from ImageNet to medical data and observed that the transfer of first-order weight statistics provided faster convergence. We show that even when taking the scaling effect into account, additional gains from second-order statistics are identified.

Other works have considered PCA-based convolutional filters either as a model by itself without training [18, 13], as an initialization [44, 49], or to estimate the dimensionality of intermediate activations [11, 37]. Note that our results suggest an initialization by *sampling* from the data covariance instead of initializing the filters directly using the principal axes of the data. Ye et al. [52] show that a "deconvolution" of data at input and intermediate layers can be beneficial. This deconvolution corresponds to a whitening of the data distribution, therefore aligning data with an isotropic weight initialization, which is related to a positive effect of alignment we observe in this paper.

In addition, there is a large body of work on unsupervised learning. Among these, the *Exemplar-CNN* method [15] can be seen as the limiting case of using random labels with infinitely many classes (one label per image) and large-scale data augmentation. In our study we do *not* use data augmentation since it provides a supervisory signal to the neural network that can cause additional effects.

## 2 Covariance matrix alignment between network parameters and data

Returning to the motivating example in Figure 1, we observe that pre-training on random labels can improve training in the downstream tasks for both random and real labels. This improvement is in the form of *faster training*. In this section, we explain this effect. We start by considering the first layer in the neural network, and extend the argument to later layers in Section 2.5.

### 2.1 Preliminaries

Let $\mathcal{D}$ be the probability distribution over the instance space $\mathcal{X} \subseteq \mathbb{R}^d$ and $\mathcal{Y}$ be a finite target set. We fix a network architecture, a loss function, a learning rate/schedule, and a distribution of weights for random initialization. Then, "training on random labels" corresponds to the following procedure: We randomly sample i.i.d. instances $\mathbf{x}_1, ..., \mathbf{x}_N \sim \mathcal{D}$, and i.i.d. labels $\mathbf{y}_1, ..., \mathbf{y}_N \in \mathcal{Y}$ independently of each other. We also sample the initial weights of the neural network, and train the network on the data $\{(\mathbf{x}_i, \mathbf{y}_i)\}_{i=1,...,N}$ for $T$ training iterations using stochastic gradient descent (SGD). During training, the weights are *random variables* due to the randomness of the initialization and the training sample. Hence, we can speak of their statistical properties, such as their first and second moments.

In the following, we are interested in layers that are convolutional or fully connected. We assume that the output of the $k$-th neuron in the first layer can be written as: $f_k(x) = g(\langle w_k, x \rangle + b_k)$ for some activation function $g$. We write $\mu_x = \mathbb{E}[\mathbf{x}]$ and observe that since the covariance matrix $\Sigma_x = \mathbb{E}[(\mathbf{x} - \mu_x) \cdot (\mathbf{x} - \mu_x)^T]$ is symmetric positive semi-definite, there exists an orthogonal decomposition $\mathbb{R}^d = V_1 \oplus ... \oplus V_r$ such that $V_i$ are eigenspaces to $\Sigma_x$ with distinct eigenvalues $\sigma_i^2$.

**Definition 1** (Alignment). *A symmetric matrix $A$ is said to be **aligned** with a symmetric matrix $B$ if each eigenspace of $B$ is a subset of an eigenspace of $A$. If $A$ is aligned with $B$, we define for each eigenvalue of $B$ with eigenspace $V \subseteq \mathbb{R}^d$ the **corresponding** eigenvalue of $A$ as the one belonging to the eigenspace that contains $V$.*

If $A$ and $B$'s eigenspaces are all of dimension 1 (which is true except for a Lebesgue null set in the space of symmetric matrices), "$A$ is aligned with $B$" becomes equivalent to the assertion that they share the same eigenvectors. However, the relation is not symmetric in general (e.g. only scalar multiples of the identity matrix $I_d$ are aligned with $I_d$, but $I_d$ is aligned with any symmetric matrix).

### 2.2 Alignment for centered Gaussian inputs

**Proposition 1.** *Assume the instances $\boldsymbol{x}$ are drawn i.i.d. from $\mathcal{N}(0, \Sigma_x)$ and the initial weights in the first layer are drawn from an isotropic distribution (e.g. the standard Gaussian). Let $\boldsymbol{w} \in \mathbb{R}^d$ be a random variable whose value is drawn uniformly at random from the set of weights in the first layer after training using SGD with random labels (see Section 2.1). Then: (1) $\mathbb{E}[\boldsymbol{w}] = 0$ and (2) $\Sigma_w = \mathbb{E}[\boldsymbol{w} \cdot \boldsymbol{w}^T]$ is aligned with the covariance matrix of data $\Sigma_x$.*

*Proof.* The proof exploits symmetries: The input, initialization, and gradient descent are invariant under the product of the orthogonal groups of the eigenspaces of $\Sigma_x$, so the distribution of weights must have the same invariance properties. The full proof is given in Appendix C. □

Proposition 1 says that independently of many settings (e.g. number of random labels, network architecture, learning rate or schedule), the eigenspaces of $\Sigma_w \in \mathbb{R}^{d \times d}$ are given by the eigenspaces of $\Sigma_x \in \mathbb{R}^{d \times d}$. Hence, the only information needed to fully determine $\Sigma_w$ is a function $f$ that maps the eigenvalues $\sigma_i^2$ of $\Sigma_x$ to the corresponding eigenvalues $\tau_i^2$ of $\Sigma_w$. Note that the argument of the proof of Proposition 1 also apply to the case of a single training run of an infinitely wide network in which the layers are given by weight vector distributions, see e.g. [47]. For finite networks, in practice, one would only be able to approximate $\Sigma_w$ based on several independent training runs.

Next, we present experimental evidence that first-layer alignment actually takes place, not just for Gaussian input with random labels, but also in real image datasets with random labels, and even when training on real labels using convolutional networks. The intuition behind this result for real labels is that small patches in the image (e.g. $3 \times 3$) are nearly independent of the labels. Before we do that, we introduce a suitable measure of alignment that we use in the experiments.

Figure 2: Plots of the misalignment scores of the filters of the first layer in a two-layer neural network (256 convolutional filters, 64 fully-connected nodes) when trained on CIFAR10 with either real or random labels. Throughout training, misalignment scores between the filters of the first layer and the data remain very small compared to those between filters and a random orthonormal basis.

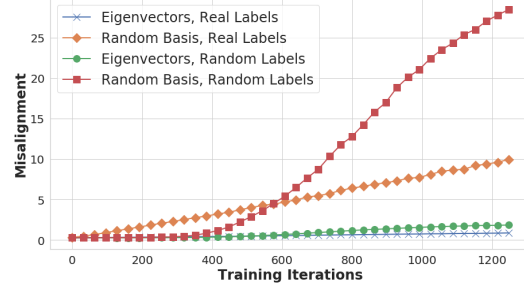

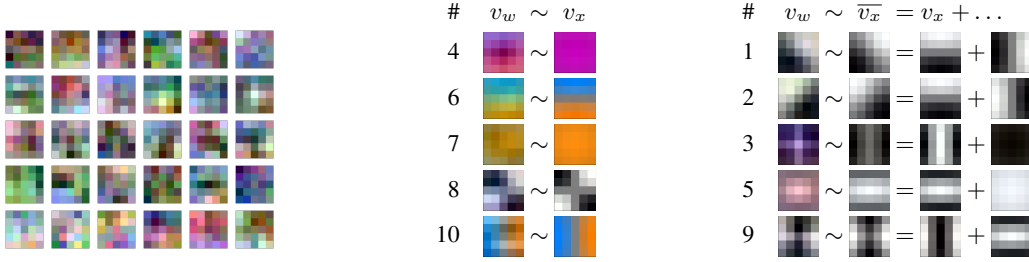

Figure 3: Visualization of covariance alignment. LEFT: Random selection of WRN-28-4 first-layer convolutional filters (CIFAR10, random labels). CENTER/RIGHT: Eigenvectors $v_w$ of $\Sigma_w$ with largest eigenvalues (rank in column '#') and eigenvectors $v_x$ of $\Sigma_x$ with $\langle v_x, v_w \rangle > 0.4$. CENTER: Cases where one $v_x$ matches. RIGHT: Cases where two $v_x$ and their weighted combination $\overline{v_x}$ match.

**Definition 2.** *For two positive definite matrices $A, B$, the "misalignment" $M(A, B)$ is defined as:*

$$M(A,B) := \inf_{\substack{\Sigma \succ 0 \text{ aligned with } A}} \left\{ \tfrac{1}{2}\boldsymbol{tr}(\Sigma^{-1}B + B^{-1}\Sigma) - d \right\} \tag{1}$$

The rationale behind this definition of misalignment is presented in Appendix D. In particular, it can be shown that for any $A, B \succ 0$, we have $M(A, B) \geq 0$ with equality if and only if $B$ is aligned with $A$. In addition, $M(A, B)$ is continuous in $B$ and satisfies desirable equivariance and invariance properties and can be computed in closed form by $M(A, B) + d = \sum_{i=1}^{r} \sqrt{\boldsymbol{tr}(B|_{V_i}) \cdot \boldsymbol{tr}(B^{-1}|_{V_i})}$ where $V_1 \oplus ... \oplus V_r$ is the orthogonal decomposition of $\mathbb{R}^d$ into eigenspaces of $A$, and $B|_{V_i}$ is the linear map $V_i \to V_i, \mathbf{v} \mapsto pr_i(B(\mathbf{v}))$ with $pr_i$ the orthogonal projection $\mathbb{R}^d \to V_i$.

Figure 2 displays the misalignment scores between the covariance of filters at the first layer with the covariance of the data (patches of images). For comparison, the misalignment scores with respect to some random orthonormal basis are plotted as well. As predicted by Proposition 1, the weight eigenvectors stay aligned to the data eigenvectors but not to an arbitrary random basis.

For image data, we can also visualize the alignment of $\Sigma_w$ to $\Sigma_x$. Figure 3 shows results based on 70 wide ResNet models [54] trained on CIFAR10 with random labels. For better visualization, we use a $5 \times 5$ initial convolution here. The left part of the figure shows a random selection of some of the $70 \cdot 64$ convolution filters. From the filters we estimate $\Sigma_w$ and compute the eigenvectors $v_w$, then visualize the ten $v_w$ with the largest eigenvalues. From the image data we compute the patch data covariance $\Sigma_x$ and its eigenvectors $v_x$. We show the data eigenvectors for which the inner product with the filter eigenvectors exceeds a threshold and the weighted sum $\overline{v_x}$ of these if there are multiple such $v_x$. (See Appendix D.1. for why this is expected to occur as well.)

The visual similarity between the $v_w$ and the $\overline{v_x}$ illustrates the predicted covariance alignment. Note that this alignment is visually non-obvious when looking at individual filters as shown on the left.

## 2.3 Mapping of eigenvalues

As stated earlier, Proposition 1 shows that, on average, the first layer effectively learns a function which maps each eigenvalue of $\Sigma_x$ to the *corresponding* eigenvalue of $\Sigma_w$ (see Definition 1). In this section, we examine the shape of this function.

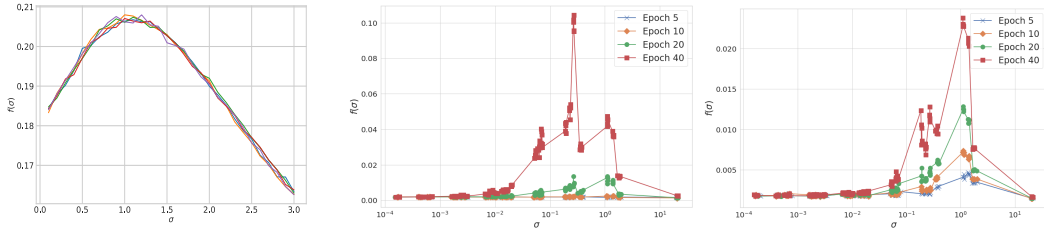

Figure 4: LEFT: $f(\sigma)$ for synthetic data $\mathcal{N}(0, \mathrm{diag}(0.1, 0.2, \ldots, 3.0))$ in fully-connected neural networks with two layers of size 256. The graph is approximately continuous and of a regular structure: increasing, then decreasing. CENTER,RIGHT: $f(\sigma)$ that results from training a 2-layer convolutional network (256 filters followed by 64 fully-connected nodes) on CIFAR10 for random (CENTER) and real labels (RIGHT) after 5, 10, 20, and 40 epochs ($\sim$195 training iterations per epoch).

Since, in practice, we will only have *approximate* alignment due to the finiteness of the number of inputs and weights, non–Gaussian inputs, and correlations between overlapping patches, we extend the definition of $f(\sigma)$. Such an extension is based on the following identity (2): For $\Sigma_x \in \mathbb{R}^{d \times d}$ let $\mathbf{v}_i$ be an eigenvector of length 1 with eigenvalue $\sigma_i^2$. If $\Sigma_w$ is aligned with $\Sigma_x$, $\mathbf{v}_i$ is also an eigenvector of $\Sigma_w$ and the corresponding eigenvalue $\tau_i^2$ is:

$$\tau_i^2 = \mathbf{v}_i^T \Sigma_w \mathbf{v}_i = \mathbf{v}_i^T \mathbb{E}[(\mathbf{w} - \mu_w)(\mathbf{w} - \mu_w)^T]\mathbf{v}_i = \mathbb{E}[\langle \mathbf{w} - \mu_w, \mathbf{v}_i \rangle^2], \tag{2}$$

which is the variance of the projection of the weight vectors onto the principal axis $\mathbf{v}_i$. We can take this as the definition of $\tau_i$ in the general case, since this formulation can be applied even when we have an imperfect alignment between the eigenspaces.

**Definition 3.** *Given two positive definite symmetric $d \times d$ matrices $\Sigma_x, \Sigma_w$, such that $\Sigma_w$ is aligned with $\Sigma_x$ or $\Sigma_x$ has $d$ distinct eigenvalues. Let $\sigma_1^2, \sigma_2^2, \ldots$ be the eigenvalues of $\Sigma_x$ with corresponding eigenvectors $\mathbf{v}_1, \mathbf{v}_2, \ldots$ of length 1, we define the transfer function from $\Sigma_x$ to $\Sigma_w$ as*

$$f : \{\sigma_1, \sigma_2, \ldots\} \to \mathbb{R}, \quad \sigma_i \mapsto \sqrt{\mathbf{v}_i^T \Sigma_w \mathbf{v}_i} \tag{3}$$

In practice, the eigenvalues are distinct almost surely so every eigenvalue of the data has a unique corresponding eigenvector of length 1 (up to $\pm$) and the function $f(\sigma)$ is well-defined.

Using this definition of $f(\sigma)$, we can now look at the shape of the function for concrete examples. Here, we train a simple fully connected network 50 times, collect statistics, and plot the corresponding mapping between each eigenvalue $\sigma_i$ of $\Sigma_x$ with the corresponding $\tau_i$ in (2) (see Appendix E.1 for more details). The result is shown in Figure 4 (LEFT). In general, $f(\sigma)$ on synthetic data looks smooth and exhibits a surprising structure: the function first increases before it decreases. Both the decreasing part or the increasing part may be missing depending on the setting (e.g. dimensionality of data and network architecture) but we observe the same shape of curves in all experiments. For real data (CIFAR10), Figure 4 (CENTER/RIGHT) shows that the function $f(\sigma)$ appears to have a similar shape (increasing, then decreasing, but less smooth) for training with both real and random labels.

We interpret (without a formal argument) this surprisingly regular shape of $f(\sigma)$ to be the result of two effects: (1) Larger eigenvalues $\sigma_i$ lead to larger effective learning rate in gradient descent, which leads in turn to larger corresponding $\tau_i$, hence the increasing part of $f$. (2) Very large eigenvalues $\tau_i$ would dominate the output of the layer, masking the contribution of other components. Backpropagation compensates for this effect to capture more of the input signal. This leads to the decreasing part of $f$ for higher $\sigma_i$. (See also Appendix E.1)

### 2.4 Covariance alignment and eigenvalue mapping explains positive transfer experimentally

To connect the alignment effect to the faster learning downstream, we conduct the following experiment. Suppose that instead of pre-training on random labels, we sample from the Gaussian approximation of the filters in the first layer that were trained on random labels. In a simple CNN on CIFAR10, consisting of one convolutional and one fully connected layers, the gain in downstream task

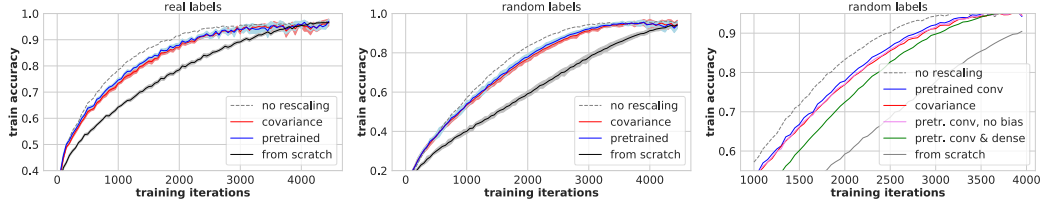

Figure 5: Training accuracy for learning real labels (LEFT) and random labels (MIDDLE, zoomed in: RIGHT) on CIFAR10 with a simple CNN (one $3 \times 3$ convolution, one hidden dense layer). Randomly initializing the convolutional filters from a learned covariance reproduces the effect of pre-training within measurement error (red and pink lines are almost indistinguishable in the right plot).

is almost fully recovered, as shown in Figure 5. The gray curves show the raw effect of pre-training, but this contains the scaling effect. To eliminate the latter effect, we always re-scale the weights after pre-training to match their $\ell_2$ norm at initialization (using $\ell_1$ norm gives similar results). Recovering the transfer effect in this way implies that the positive transfer is mainly due to the second-order statistics of the weights in the first layer, which, by Proposition 1, are fully described by the alignment of principal components in combination with the shape of the function $f(\sigma)$.

Note that the combined results presented up to here indicate that both analytically and experimentally the following seems to be true: Given the training data of a neural network that is trained with random labels, we can predict the second-order statistics of its first layer weights up to a one-dimensional scaling function $f(\sigma)$ of the eigenvalues and this function has a surprisingly regular shape. If further investigation of $f$ may lead us to understand its shape (which we regard as an interesting area of research), we could predict these after-training statistics perfectly by only gathering the data statistics.

## 2.5 Deeper layers

So far, we have only discussed effects in the first layer of a neural network. However, in Figure 6 we show that transferring more of the layers from the model pre-trained on random labels improves the effect considerably. We now turn to generalizing Section 2.4 to the multi-layer case, i.e. we reproduce this effect with weight initializations computed from the input distribution.

For the first layer, we have seen that we can reproduce the effect of training on random labels by randomly sampling weights according to the corresponding covariance matrix $\Sigma_w$, which in turn is given by the same eigenvectors $e_1, ..., e_d$ as the data covariance $\Sigma_x$, and a set of new eigenvalues $\tau_1^2, ..., \tau_d^2$. So, if we can approximate the right (or good) eigenvalues, we can directly compute an initialization that results in faster training in a subsequent task of learning real labels. See the first two accuracy columns in Table 1 for results in an example case, and Appendix E for different choices of $\tau_1, ..., \tau_d$ (it turns out different reasonable choices of the $\tau_i$ give all results very similar to Table 1).

We can then iterate this procedure also for the next (fully connected or convolutional) layers. Given the filters for the earlier layers $L_1, ..., L_{k-1}$, for each training image we can compute the output after layer $L_{k-1}$, which becomes the input to the layer $L_k$. Treating this as our input data, we determine the eigenvectors $e_1, e_2, ...$ of the corresponding data covariance matrix. Then we compute the $d$ most important directions and use $\tau_1 e_1, \tau_2 e_2, ..., \tau_d e_d$ (with the same assumed $\tau_1, \tau_2, ...$ as before) as our constructed filters. (Alternatively, we can sample according to the covariance matrix given by the eigenvectors $e_i$ and eigenvalues $\tau_1^2, ..., \tau_d^2, 0, ..., 0$, which gives again essentially the same results, compare Table 2 and 4 in Appendix E.3.)

Applying this recipe to a CNN with three convolutional layers and one fully connected layer, we see that this indeed gives initializations that become better when applied to 1,2, and 3 layers, see Table 1. The performance after applying this approach to all three convolutional layers matches the performance of transferring the first three layers of a network trained on random labels. See Appendix E.3 for details.

Table 1: Training and test accuracy on subsets of CIFAR10 of the initialization procedure described in Section 2.5 on the layers of a simple convolutional network. Both training and test accuracies improve with the number of layers that are initialized in this way.

| Iterations | Data | Convolutional layers sampled | | | |
| | | $\{\}$ | $\{1\}$ | $\{1,2\}$ | $\{1,2,3\}$ |
|---|---|---|---|---|---|
| 100 | Train | 0.31 | 0.34 | 0.38 | 0.41 |
| | Test | 0.31 | 0.33 | 0.37 | 0.40 |
| 1000 | Train | 0.58 | 0.61 | 0.67 | 0.68 |
| | Test | 0.53 | 0.55 | 0.56 | 0.56 |

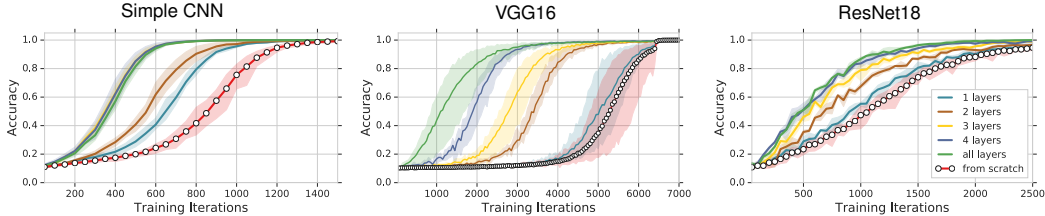

Figure 6: Transferring more layers improves downstream performance. Simple CNN architecture with 3 convolutional layers (LEFT), VGG16 (CENTER), and ResNet18 (RIGHT) pre-trained on CIFAR10 examples with random labels and subsequently fined-tuned on 25k fresh CIFAR10 examples with random labels. Lines with circular markers correspond to training from scratch. Error bars correspond to min/max over 3 runs. Plots for fine-tuning with real labels available in the appendix.

# 3    Specializing neurons

Despite the alignment effect taking place at the earlier layers of the neural network when trained with random labels, negative transfer is sometimes observed when fine-tuning on a downstream task as shown in Figure 1. In this section, we show that this is likely due to a specialization at the later layer.

Figure 7 displays the distribution of neurons with respect to the number of held out images they are activated by for the settings of Figure 1 that exhibited positive (top row) and negative (bottom row) transfers. Comparing neural activations during initialization, end of pre-training, and end of fine-tuning, we note that neural activations are markedly diminished in the negative transfer case compared to the positive transfer case despite the fact that their neural activation distributions were identical during initialization. In Appendix F, we show that the significant drop in neural activation in the negative transfer case happens immediately after switching to the downstream task. As a result, the effective capacity available downstream is diminished. By contrast, neural activations are not severely impacted in the positive transfer setting. In Appendix F, we provide detailed figures describing this phenomenon across all layers of VGG16, which reveal that such a specialization effect becomes more prominent in the later layers compared to the earlier layers. In particular, Appendix F shows that neural activations at the later layers can drop abruptly and permanently once the switch to the downstream task takes place due to specialization, which can prevent the network for recovering its fully capacity.

One way to mitigate the effect of the inactive ReLU units is to increase the width so that the capacity remains sufficiently large for the downstream task. Figure 8 shows that increasing the width can indeed mitigate the negative transfer effect. While increased width seems to have general performance advantages [54], it seems to be also particularly useful in the case of transfer learning [30].

# 4    Concluding remarks

The objective of this paper is to answer the question of what neural networks learn when trained on random labels. We provide a partial answer by proving an alignment effect of principal components of network parameters and data and studying its implications, particularly for transfer learning. One important consequence is that second-order statistics of the earlier layers can be reduced to a one-dimensional function, which exhibits a surprising, regular structure. It remains an open question what the "optimal" shape of such function is, or whether it can be described analytically.

The models used in this paper are taken from a large set of experiments that we conducted using popular network architectures and datasets, such as simple convolutional networks, VGG16, ResNet18-v2,

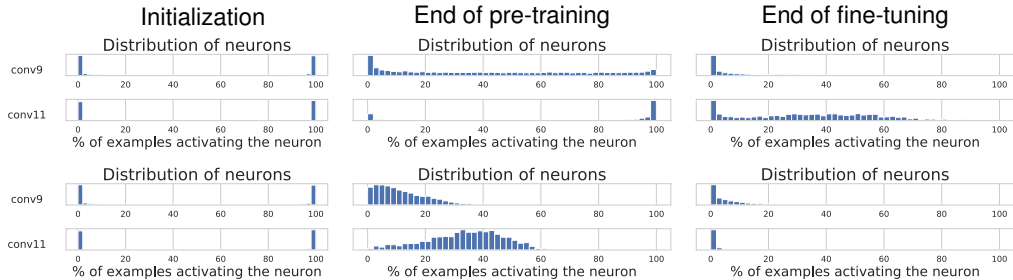

Figure 7: Activation plots for the two VGG16 models in Figure 1 at initialization (LEFT), after pre-training with random labels (CENTER), and after subsequently fine-tuning on fresh examples with random labels (RIGHT). Top row is for the positive transfer case; bottom row shows negative transfer. Histograms depict distributions of neurons over the fraction of *held out* examples that activate them. The two histograms in each subplot correspond to two different activation spaces.

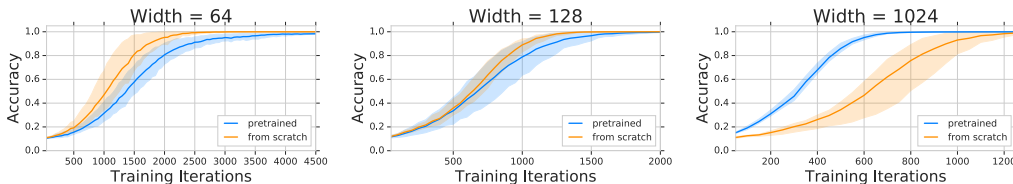

Figure 8: Increasing model width mitigates negative transfer. Simple CNN architectures with two convolutional layers and 64 (LEFT), 128 (CENTER), and 1024 (RIGHT) units in the dense layer.

CIFAR10, and ImageNet, with wide range of hyperparameter settings (Appendix B). These experiments show that pre-training on random labels very often accelerates training on downstream tasks compared to training from scratch with the *same hyperparameters* and rarely hurts the training speed.

By studying what is learned on random labels, we shed new insights into previous phenomena that have been reported in the literature. For instance, the alignment effect at the earlier layers explains the empirical observations of [42] that pre-training on random labels can accelerate training in downstream tasks and the observation of [4] that the filters learned on random labels seem to exhibit some useful structure. Also, our findings related to the inactive ReLU units at the later layers demonstrate how upper layers specialize early during training, which may explain why neural networks exhibit critical learning stages [1] and why increasing the width seems to be particularly useful in transfer learning [30]. Both alignment and specialization are in agreement with the observation that earlier layers generalize while later layers specialize, a conclusion that has been consistently observed in the literature when training on real labels [3, 4, 10, 53]. We show that it holds for random labels as well.

## Acknowledgements

The authors are grateful to Alexander Kolesnikov, Alexandru Țifrea, Jessica Yung, Larisa Markeeva, Lionel Ngoupeyou Tondji, Lucas Beyer, Philippe Gervais, and other members of the Google Brain Team for valuable discussions.

## Broader Impact

This work is partially theoretical and contains experiments to study the theoretical results and related hypotheses. The paper aims at improving our understanding of how DNNs learn from data and therefore does not have a *direct* impact on applications or society. Hence, speculating on its potential broader impact is difficult at this stage. Nevertheless, we hope that a better understanding of deep neural networks will lead to improvements in the future along the direction of building interpretable and explainable AI, which are critical ingredients for the creation of socially-responsible AI systems.

## Funding Disclosure

This work was performed at and funded by Google.

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
