[Supplementary Material]

# A    Experimental details

## A.1    Upstream and downstream datasets

We use two datasets: CIFAR10 [31] and ImageNet ILSVRC-2012 [14].

For each run of a transfer experiment we

1. randomly sample `num_examples_upstream` examples uniformly from the training split of the dataset to form the (upstream) training set for pre-training;

2. randomly sample `num_examples_downstream` examples from the remainder of the training split of the dataset. They form the (downstream) training set for fine-tuning. This guarantees that the upstream and downstream examples do not intersect;

3. upstream training examples are labelled with `num_classes_upstream` classes randomly and uniformly;

4. when fine-tuning on random labels, we randomly and uniformly label the downstream examples with `num_classes_downstream` classes.

## A.2    Neural network architectures

Throughout the main text we use three different architectures: a simple convolutional architecture "Simple CNN", VGG16, and ResNet18. The Simple CNN can be further configured by specifying the number of convolutional layers and units in the dense layer. We use the same architectures for both CIFAR10 and ImageNet datasets, by only adjusting the number of outputs (logits).

**Simple CNN** is a convolutional architecture consisting of:

1. `num_conv_layers` convolutional layers with $3 \times 3$ filters, each followed by the ReLU activation. Each convolutional layer contains `num_filters` filters (with biases) that are applied with stride 1.

2. The outputs of the final convolutional layer are flattened and passed to a dense layer (with biases) of `num_units` units, followed by the ReLU activation.

3. The classifier head, i.e. a dense layer with `num_output` units (logits).

The **VGG16** architecture that we use is "Configuration D" from Table 1 of [46] with two dense layers ("FC-4096") removed:

```
conv0:   64 filters
conv1:   64 filters
maxpool
conv2:  128 filters
conv3:  128 filters
maxpool
conv4:  256 filters
conv5:  256 filters
conv6:  256 filters
maxpool
conv7:  512 filters
conv8:  512 filters
conv9:  512 filters
maxpool
conv10: 512 filters
conv11: 512 filters
conv12: 512 filters
maxpool
dense layer with num_outputs units (classifier head)
```

All convolutional filters (with biases) are of size $3 \times 3$ and applied with `SAME` padding and stride 1. ReLU activation is applied after every convolutional layer. Max-pooling is performed over a $2 \times 2$ pixel window, with SAME padding and stride 2.

**ResNet18** We use vanilla ResNet-v2 architecture [22] with batch normalization [26] replaced by the group normalization [51].

## A.3 Training

All biases in the models are initialized with zeros, while all the other parameters (convolutional filters and weight matrices of the dense layers) are initialized using He normal algorithm [21] with `init_scale` scaling.

Model outputs (logits) are passed through the softmax function and we minimize the cross-entropy (i.e. the negative log-likelihood) during training. We use SGD with momentum 0.9 and batch size 256 to train our models. We start training with the specified `learning_rate` and divide it by 3 two times during the training: after $1/3 \times$`total_steps` and $2/3 \times$`total_steps` steps.

When training the models we report the accuracy on *the entire training set*, not on the mini-batch.

We do not use data augmentation in our experiments. We do not use weight decay, dropout, or any other regularization. For CIFAR10 we scale the inputs to the $[-1, 1]$ range. For ImageNet we resize the examples, take a $224 \times 224$ central crop, and scale inputs to the $[-1, 1]$ range.

## A.4 Transferring the model from the upstream to the downstream task

After the upstream pre-training finishes, we replace the classifier head with a freshly initialized one.

Unless otherwise stated, for Simple CNN and VGG16 architectures we also re-scale the model parameters after the pre-training. We store the per-layer parameter $\ell_2$ norms at the initialization and re-scale each layer of the trained model to match the stored $\ell_2$ norms. Re-scaling does not affect the classifier accuracy and predictions, since it reduces to multiplying all parameters of a given layer by a strictly positive constant. However, re-scaling changes the logits and, therefore, the cross-entropy loss. It is not immediately obvious how to re-scale the residual architecture without changing its predictions so we decided not to re-scale ResNet18.

When fine-tuning downstream we use the same initial learning rate and schedule as in upstream pre-training.

## A.5 Figure 1: Positive and negative transfer with VGG16 on CIFAR10

Experiments 1 and 2 use:

```
init_scale = 0.526
learning_rate = 0.01
num_classes_upstream = 5
num_examples_upstream = 20000
epochs_upstream = 120
```

Experiments 3 and 4 use:

```
init_scale = 0.612
learning_rate = 0.009
num_classes_upstream = 50
num_examples_upstream = 20000
epochs_upstream = 80
```

Error bars correspond to $\pm 1$ std. over 12 independent runs.

## A.6 Figure 2: Misalignment plots

To measure misalignment, we use the misalignment score in Definition 1 using the closed-from expression provided in Section 2.2. First, the eigenvectors of the $3 \times 3$ patches of images are computed, which reside in a space of dimension 27 due to the three input channels. Let $v_i$ be the eigenvectors of

data, which have distinct eigenvalues almost surely, the alignment score is then:

$$\sum_{i=1}^{d} \sqrt{(v_i^T \Sigma_w v_i) \cdot (v_i^T \Sigma_w^{-1} v_i)} - 1,$$

where the summation is taken over all 27 eigenvectors. To gain an intuition behind this formula, note that if $v_i$ is itself an eigenvector of $\Sigma_w$, then $(v_i^T \Sigma_w v_i) \cdot (v_i^T \Sigma_w^{-1} v_i) = 1$. The covariance of weights $\Sigma_w$ is estimated in a single run by computing the covariance of the filters in the first layer.

To ensure that alignment depends indeed on the eigenvectors of data, we also compute alignment in which the set of data eigenvectors $\{v_i\}$ is replaced by some random orthonormal basis of the plane. The network architecture is a 2-layer CNN: convolutional followed by a fully-connected layer. The experiment setup used to produce Figure 2 is:

```
num_conv_layers = 1
num_filters = 256
num_units = 64
learning_rate = 0.01
num_classes = 10
num_examples = 50000
epochs_upstream = 40
```

### A.7 Figure 3: ResNet convolutional filter alignment

Figure 3 was produced from 70 runs of a wide ResNet [54] on CIFAR10 with random labels. We used 4 blocks per group and a width factor of 4. As mentioned in the main text, we replaced the initial $3 \times 3$ convolution with a $5 \times 5$ convolution for better visualization. Training was run with batch normalization [26] and the following parameters:

```
init_scale = 0.01
learning_rate = 0.005
num_examples = 50000
epochs = 1800
```

### A.8 Figure 4: Plots of the function $f(\sigma)$

These figures were generated using the procedure described in Section 2.3. For every eigenvalue $\sigma_i^2$ of the data with eigenvector $v_i$, the corresponding $\tau_i^2$ is computed using Eq. (2). The pairs $(\sigma_i^2, \tau_i^2)$ define a mapping from $\mathbb{R}^+$ to $\mathbb{R}^+$, which is plotted in Figure 4.

The network architecture is a 2-layer CNN: convolutional followed by a fully-connected layer. The experiment setup used to produce Figure 2 is:

```
num_conv_layers = 1
num_filters = 256
num_units = 64
learning_rate = 0.01
num_classes = 10
num_examples = 50000
epochs_upstream = 40
```

### A.9 Figure 5: Explaining the positive transfer

Network: SimpleCNN

```
num_conv_layers = 1
num_filters = 64
num_units = 256
```

Figure 9: Transferring more layers improves the downstream performance. Simple CNN architecture with 3 conv. layers (LEFT), VGG16 (CENTER), and ResNet18 (RIGHT) pre-trained on CIFAR10 examples with random labels and subsequently fined-tuned on 25k fresh CIFAR10 examples with real labels (TOP) and 10 random labels (BOTTOM). Lines with circular markers correspond to training from scratch. Error bars correspond to min/max over 3 independent runs.

Training:

```
num_examples_upstream = 10000
num_classes_upstream = 10
num_examples_downstream = 10000
num_classes_downstream = 10
init_scaling = 1.0
learning_rate = 0.0005
total_steps = 10000
```

For all curves with the exception of the "no rescaling" curve, each layer is scaled down by a factor between pre-training and training to match the $\ell_2$ norm of the weights at initialization.

For all runs, the head layer is re-initialized after pre-training. For "pretrained conv" and "pretrained conv, no bias" also the fully connected layer is re-initialized. For "pretrained conv, no bias" also the bias of the convolutional layer is reset to zero. This is the fairest comparison to "covariance": For "covariance" the filters in the convolutional layer are random samples from a Gaussian distribution with mean 0 and the covariance obtained from training on random labels. The bias is set to zero and the dense and head layers are initialized randomly.

Each line is the average of 12 runs that differ in the random initializations. In the left and center image, the "covariance", "pretrained", and "from scratch" curves are surrounded by a colored area indicating $\pm 1$ standard deviations. The "no rescaling" curve is without this area, it would touch the "pretrained" curve below it. In the right image no error bounds are plotted since the curves are too close together.

### A.10 Figure 6: Transferring more layers improves the downstream performance

Figure 9 is the extended version of Figure 6 that includes the models fine-tuned on real labels (apart from the models fine-tuned on the random labels).

The left column (Simple CNN architecture) uses:

```
init_scale = 0.518
learning_rate = 0.01
num_classes_upstream = 25
num_examples_upstream = 15000
epochs_upstream = 40
num_conv_layers = 3
num_filters = 16
```

```
num_units = 1024
```

The center column (VGG16) corresponds to the same setup as in Experiments 1 and 2 in Figure 1. It uses:

```
init_scale = 0.526
learning_rate = 0.01
num_classes_upstream = 5
num_examples_upstream = 20000
epochs_upstream = 120
```

The right column (ResNet18) uses:

```
init_scale = 0.671
learning_rate = 0.01
num_classes_upstream = 50
num_examples_upstream = 25000
epochs_upstream = 80
```

## A.11 Figure 7: Neuron activation plots

Figures 10 and 11 are the extended versions of Figure 7. They include activation plots for *all intermediate layers* of VGG16 models (i) at initialization, (ii) in the end of the pre-training, (iii) in the end of fine-tuning on real labels, (iv) in the end of fine-tuning on 10 random labels. Figures 10 and 11 illustrate the negative and positive transfer examples respectively.

The VGG16 model from Figure 10 (top row in Figure 8) uses the same setup as in Experiments 3 and 4 in Figure 1:

```
init_scale = 0.612
learning_rate = 0.01
num_classes_upstream = 50
num_examples_upstream = 20000
epochs_upstream = 80
```

The VGG16 model from Figure 11 (bottom row in Figure 8) uses the same setup as in Experiments 1 and 2 in Figure 1:

```
init_scale = 0.526
learning_rate = 0.01
num_classes_upstream = 5
num_examples_upstream = 20000
epochs_upstream = 120
```

## A.12 Figure 8: Increasing the width mitigates the negative transfer

The models from all three subplots use Simple CNN architecture and share the same parameters:

```
init_scale = 1.218
learning_rate = 0.012
num_classes_upstream = 2
num_examples_upstream = 10000
epochs_upstream = 100
num_conv_layers = 2
num_filters = 16
```

The models from LEFT, CENTER, and RIGHT subplots use num_units of 64, 128, and 1 024, respectively. Error bars correspond to max/min over 3 independent runs.

Figure 10: Neuron activations in case of the negative transfer. The VGG16 model pre-trained for 1 (TOP-LEFT) and 6240 (TOP-RIGHT) training iterations on 20k examples from CIFAR10 with 50 random classes and subsequently fine-tuned for 200 epochs on fresh 25k examples from CIFAR10 with real labels (BOTTOM-LEFT) and 10 random labels (BOTTOM-RIGHT). In each subplot, the left column depicts distributions of neurons over the fraction of input examples that activate them. The right column depicts distribution of the input examples over the fraction of neurons that are activated by them. The 5k input examples were taken from the holdout test split of CIFAR10.

Figure 11: Neuron activations in case of the positive transfer. The VGG16 model pre-trained for 1 (TOP-LEFT) and 9360 (TOP-RIGHT) training iterations on 20k examples from CIFAR10 with 5 random classes and subsequently fine-tuned for 200 epochs on fresh 25k examples from CIFAR10 with real labels (BOTTOM-LEFT) and 10 random labels (BOTTOM-RIGHT). In each subplot, the left column depicts distributions of neurons over the fraction of input examples that activate them. The right column depicts distribution of the input examples over the fraction of neurons that are activated by them. The 5k input examples were taken from the holdout test split of CIFAR10.

# B Empirical evidence with diverse real-world settings

We argued theoretically that the alignment effect holds under certain idealized conditions (Proposition 1) and demonstrated that it entirely explains the positive transfer in a real-world setting (Figure 5). In this appendix, we confirm that the positive transfer is reproducible and can be frequently observed in common real-world settings with various popular network architectures, datasets, and different hyperparameters.

**Experimental setup** We run multiple transfer experiments, repeating each run with several different seeds. We consider three network architectures: (a) a simple convolutional architecture with a configurable number of conv. layers, filters, and units in one dense layer, (b) VGG16 [46] with two final dense layers of width 4096 removed, and (c) ResNet-v2 architecture [22]. The *disjoint* upstream and downstream training sets are sampled randomly from one of the two datasets: CIFAR10 [31] or ImageNet ILSVRC-2012 [14]. We never transfer *between different* datasets. The models are pre-trained with random labels and fine-tuned for a fixed number of epochs with either real or random labels using SGD with momentum 0.9. We randomly explore various initial learning rates, initialization types and scales, numbers of random classes and examples upstream, duration of the pre-training, and configurations of the Simple CNN architecture.

We collect two sets of experiments for CIFAR10 (Experiments A and B) and two sets for ImageNet (Experiments A and B) reported below. Each set consists of multiple *groups* of experiments. Experiments are gathered in a *group* if they share same (1) architecture, (2) learning rate, (3) initialization type and scale, (4) number of examples upstream, (5) number of classes upstream, and (6) number of epochs upstream. For each configuration of these 6 parameters we explore *all possible combinations* of the following choices: (a) the random seed, (b) pre-train / train from scratch on downstream, (c) transfer one layer / all layers [optional], (d) train downstream with real / random labels. Gathering experiments in groups like this allows to compare the downstream performance of the pre-trained models to that of models trained from scratch *with same hyperparameters*.

**Visualizing the experiments** In order to visualize the experiments we summarize each group with two numbers: one characterizing the downstream performance of the pre-trained models, and one for the models trained from scratch. In order to capture the speed of training we use the *area under the curve* (AUC) to sketch the training with a single number. For instance, the pre-trained model in Experiment 2 of Figure 1 (blue line) trains faster than the one trained from scratch (orange line). Accordingly, the area under the blue curve is larger than the area under the orange curve. We can now visualize all the groups of experiments on a single scatter plot. Depending on what exact curves we summarize with AUC, we get three scatter plots: (1) training accuracy when using real labels downstream (areas under the solid lines in Experiments 1 and 3 in Figure 1), (2) test accuracy when using real labels downstream (areas under the dotted lines in Experiments 1 and 3 in Figure 1), (3) training accuracy when using random labels downstream (areas under the solid lines in Experiments 2 and 4 in Figure 1).

## B.1 CIFAR10, Experiments A

This set of experiments counts 20 groups per architecture. We use 2 different random seeds, resulting in 16 experiments per group, or $3 \times 20 \times 16 = 960$ trained models in total.

The following parameters are shared between all runs:

```
num_examples_downstream = 25000
num_epochs_downstream = 200
init_algorithm = "he"
```

For each group of experiments we sample the following parameters randomly:

```
init_scale = random log_uniform(0.5, 1.35)
learning_rate = random log_uniform(0.008, 0.0125)
num_examples_upstream = random uniform([5000, 10000, 15000, 20000, 25000])
num_classes_upstream = random uniform([2, 5, 10, 25, 50, 100])
num_epochs_upstream = random uniform([20, 40, 80, 100, 120, 150, 200])
```

For the Simple CNN architecture we also randomly sample

```
num_conv_layers = random uniform([1, 2, 3, 4, 5])
num_units = random uniform([64, 128, 256, 512, 1024])
num_filters = random uniform([16, 32])
```

Exmeriments A are depicted on Figure 12. Two clusters of points in the CENTER row correspond to models with VGG16 architecture ($y > 0.7$) and models with SimpleCNN or ResNet18 architecture ($y \leq 0.7$). Surprisingly, several experiments with transferring one (first) layer of VGG16 when fine-tuning with real labels lead to improved test accuracy compared to the same models trained from scratch (blue squares, CENTER row).

## B.2   CIFAR10, Experiments B

This set of experiments contains 20 groups per architecture. We use 2 different random seeds, resulting in 16 experiments per group, or $3 \times 20 \times 16 = 960$ trained models in total.

We use slightly wider hyperparameter ranges compared to the CIFAR10 Experiments A. Also, we include the orthogonal initialization algorithm [45]. The following parameters are shared between all runs:

```
num_examples_downstream = 25000
num_epochs_downstream = 200
```

For each group of experiments we sample the following parameters randomly:

```
init_algorithm = random uniform(["he", "orthogonal"])
init_scale = random log_uniform(0.5, 2.)
learning_rate = random log_uniform(0.005, 0.02)
num_examples_upstream = random uniform([5000, 10000, 15000, 20000, 25000])
num_classes_upstream = random uniform([2, 5, 10, 25, 50, 100])
num_epochs_upstream = random uniform([20, 40, 80, 100, 120, 150, 200])
```

For the Simple CNN architecture we also randomly sample

```
num_conv_layers = random uniform([1, 2, 3, 4, 5])
num_units = random uniform([512, 1024, 2048])
num_filters = random uniform([16, 32])
```

Exmeriments B are depicted on Figure 13. The models with $y > 0.7$ on CENTER row again correspond to the VGG16 architecture. This time we do not observe improvements in the test performance when pre-training.

## B.3   ImageNet, Experiments A

This set of experiments counts 10 groups per architecture. We use 5 different random seeds. We always transfer all layers. This results in 20 experiments per group, or $3 \times 10 \times 20 = 600$ trained models in total.

The following parameters are shared between all runs:

```
num_examples_downstream = 200000
num_epochs_downstream = 100
num_epochs_upstream = 100
num_classes_upstream = 1000
init_algorithm = "he"
```

For each group of experiments we sample the following parameters randomly:

```
init_scale = random log_uniform(0.5, 1.5)
learning_rate = random log_uniform(0.008, 0.0125)
num_examples_upstream = random uniform([50000, 100000])
```

For the Simple CNN architecture we also randomly sample

```
num_conv_layers = random uniform([1, 2, 3])
num_units = random uniform([32, 64, 128, 256, 512, 1024])
num_filters = random uniform([16, 32, 64])
```

ImageNet Exmeriments A are depicted on Figure 14. All the ResNet18 runs result in positive transfer in terms of the downstream training performance, i.e. all the green points on TOP and BOTTOM plots are below the diagonal. This may be caused by the fact that we do not re-scale ResNet18 models after pre-training, which means the effective downstream learning rate of the pre-trained ResNets18 may be affected.

## B.4   ImageNet, Experiments B

This set of experiments counts 10 groups per architecture. We use 5 different random seeds. We always transfer all layers. This results in 20 experiments per group, or $3 \times 10 \times 20 = 600$ trained models in total.

We use smaller upstream and downstream training sizes compared to ImageNet Experiments A.

The following parameters are shared between all runs:

```
num_examples_downstream = 50000
num_epochs_downstream = 100
num_epochs_upstream = 100
num_classes_upstream = 1000
init_algorithm = "he"
```

For each group of experiments we sample the following parameters randomly:

```
init_scale = random log_uniform(0.5, 1.5)
learning_rate = random log_uniform(0.008, 0.0125)
num_examples_upstream = random uniform([25000, 50000])
```

For the Simple CNN architecture we also randomly sample

```
num_conv_layers = random uniform([1, 2, 3])
num_units = random uniform([32, 64, 128, 256, 512, 1024])
num_filters = random uniform([16, 32, 64])
```

ImageNet Exmeriments B are depicted on Figure 15. We see again that all the ResNet18 runs result in positive transfer in terms of the downstream training performance. However, this time we do not observe the positive transfer with architectures other than ResNet18.

Figure 12: Scatter plots for CIFAR10 Experiments A. Models fine-tuned on real labels (TOP and CENTER) and on random labels (BOTTOM). Each dot corresponds to one *group of experiments*. Points below the orange $x = y$ line on TOP correspond to experiments where models pre-trained with random labels train faster downstream (with real labels) compared to models trained from scratch *with same hyperparameters*. CENTER AND RIGHT columns contain zoomed in versions of the plots from LEFT column.

Figure 13: Scatter plots for CIFAR10 Experiments B. Models fine-tuned on real labels (TOP and CENTER) and on random labels (BOTTOM). Each dot corresponds to one *group of experiments*. Points below the orange $x = y$ line on TOP correspond to experiments where models pre-trained with random labels train faster downstream (with real labels) compared to models trained from scratch *with same hyperparameters*. CENTER AND RIGHT columns contain zoomed in versions of the plots from LEFT column.

Figure 14: Scatter plots for ImageNet Experiments A. Models fine-tuned on real labels (TOP and CENTER) and on random labels (BOTTOM). Each dot corresponds to one *group of experiments*. Points below the orange $x = y$ line on TOP correspond to experiments where models pre-trained with random labels train faster downstream (with real labels) compared to models trained from scratch *with same hyperparameters*. RIGHT column contains zoomed in versions of the plots from LEFT column.

Figure 15: Scatter plots for ImageNet Experiments B. Models fine-tuned on real labels (TOP and CENTER) and on random labels (BOTTOM). Each dot corresponds to one *group of experiments*. Points below the orange $x = y$ line on TOP correspond to experiments where models pre-trained with random labels train faster downstream (with real labels) compared to models trained from scratch *with same hyperparameters*. RIGHT column contains zoomed in versions of the plots from LEFT column.

## B.5  Summary of the experiments

Analysis of experiments presented in the previous four sections reveals a large number of positive transfers where the pre-trained model *trains faster* on the downstream task compared to the model trained from scratch using the *same hyperparameters*. Generally it looks like transferring more layers leads to stronger positive effects. All ResNet18 runs performed on ImageNet (and most of the ones on CIFAR10) result in positive transfer (in terms of the downstream training accuracy), which may be caused by the fact that we do not re-scale them after pre-training.

Most likely there are other important factors, yet to be discovered, playing a role in all these experiments. However, the evidence presented earlier (Figures 2 and 3, Table 1) suggests that alignment is responsible for the observed positive transfer in at least some of these cases.

## B.6  Detailed results for some of the experiments

Figure 16 reports experiments with ResNet18 on ImageNet where pre-training helps. These experiments differ only in the in initialization scale and otherwise use:

```
learning_rate = 0.01
num_classes_upstream = 1000
num_examples_upstream = 500000
num_epochs_upstream = 100
num_examples_downstream = 500000
num_epochs_downstream = 100
```

Figure 17 reports experiments with VGG16 on ImageNet where pre-training both helps and hurts. It may speed up the training, or make it slower. Surprisingly, in some cases it may slightly improve the holdout test accuracy. These experiments use:

```
init_scale = 1.0
learning_rate = 0.01
num_classes_upstream = 1000
num_examples_upstream = 500000
num_epochs_upstream = 100
num_examples_downstream = 500000
num_epochs_downstream = 100
```

Figure 18 reports experiments with VGG16 on ImageNet where pre-training hurts both real and random label downstream tasks. These experiment are taken from the ImageNet Experiments A reported in Section B.3. These experiments use:

```
init_scale = 0.747
learning_rate = 0.008
num_classes_upstream = 1000
num_examples_upstream = 50000
num_epochs_upstream = 100
num_examples_downstream = 200000
num_epochs_downstream = 100
```

Figure 16: Pre-training on random labels often accelerates the downstream training compared to training from scratch with same hyperparameters. ResNet18 models are pre-trained on 500k ImageNet examples with 1000 random labels and subsequently fine-tuned on the fresh 500k ImageNet examples with either real labels or 1000 random labels using learning rate 0.01 and different initialization scales. Error bars correspond to $\pm 1$ std. over 2 independent runs.

Figure 17: Pre-training on random labels may both accelerate or slow down the downstream training compared to training from scratch with same hyperparameters. VGG16 models are pre-trained on 500k ImageNet examples with 1000 random labels and subsequently fine-tuned on the fresh 500k ImageNet examples with either real labels or 1000 random labels. LEFT: Surprisingly, pre-training improves the holdout test accuracy. Error bars correspond to $\pm 1$ std. based on 2 independent runs.

Figure 18: VGG16 models are pre-trained on 50k ImageNet examples with 1000 random labels and subsequently fine-tuned on 200k fresh ImageNet examples with real labels or 1000 random labels.

## C   Proof of Proposition 1

Proposition 1 makes the following assumptions:

1. The first layer is either
   - fully connected, and the input from $\mathbb{R}^d$ is normally distributed with mean $\mu_x = 0$ and covariance $\Sigma_x$, or
   - convolutional, with patches that are not overlapping and the data in each position is independent normally distributed with mean $\mu_x = 0$ and covariance $\Sigma_x$.

2. The first layer weights $w \in \mathbb{R}^d$ are initialized i.i.d. at random from some distribution on $\mathbb{R}^d$ that is invariant under the orthogonal group $O(d)$.

3. The sampled inputs are labelled randomly according to some distribution over the target set $\mathcal{Y} = \{1, 2, \ldots, c\}$, independently of the input sample.

Let $\mathcal{G} := \{G \in O(d) \,|\, G^T \Sigma_x G = \Sigma_x\}$ be the group of orthogonal matrices that leave the distribution of the data $x_i$ invariant. Proposition 1 follows from these two claims:

**Claim 1.** *A probability distribution $\mathcal{D}_w$ on $\mathbb{R}^d$ with mean $\mu_w$ and covariance matrix $\Sigma_w$ that is invariant under $\mathcal{G}$ has $\mu_w = 0$ and $\Sigma_w$ aligned with $\Sigma_x$.*

**Claim 2.** *The probability distribution of the weights $w$ in the first layer after $t$ iterations of training is invariant under $\mathcal{G}$ (for any $t = 0, 1, \ldots$).*

*Proof of Claim 1.* Since $-I \in \mathcal{G}$, we have $\mathbb{E}[w] = \mathbb{E}[-w]$, so we must have $\mathbb{E}[w] = 0$. For the second part, assume

$$\mathbb{R}^d = V_1 \oplus V_2 \oplus \ldots \oplus V_r$$

is the orthogonal decomposition of $\mathbb{R}^d$ into eigenspaces of $\Sigma_x$ and $d_i := \dim(V_i)$ are the dimensions of its parts. Then

$$\mathcal{G} = O(d_1) \times O(d_2) \times \ldots \times O(d_r)$$

where each $O(d_i)$ operates in the canonical way on $V_i$ and leaves the other parts invariant.

By definition of $\mathcal{G}$ we have $G^T \Sigma_w G = \Sigma_w$ for all $G \in \mathcal{G}$. Then each $G \in \mathcal{G}$ must also leave eigenspaces of $\Sigma_w$ invariant since for an eigenvector $u$ with $\Sigma_w u = \lambda u$, we have

$$\Sigma_w(Gu) = (\Sigma_w G)u = (G\Sigma_w)u = \lambda \cdot Gu.$$

We have to show that each $V_i$ is contained in an eigenspace of $\Sigma_w$. Assume to the contrary that a particular $V_i$ is not contained in any eigenspace of $\Sigma_w$. Since the eigenvectors of $\Sigma_w$ span $\mathbb{R}^d$, there must be an eigenvector $u$ in an eigenspace $U$ of $\Sigma_w$ that is not orthogonal to $V_i$. We will show that $V_i \subseteq U$: Let $M_i \in \mathcal{G}$ be the matrix that is $I$ on $V_i$ and $-I$ on all other $V_j$, $j \neq i$, and let $pr_i : \mathbb{R}^d \to V_i$ be the orthogonal projection onto $V_i$. Then by assumption $pr_i(u) \neq 0$, but since $pr_i(u) = (u + M_i u)/2$, we must also have $pr_i(u) \in U$. Since $O(d_i)$ operates transitively on the set of all vectors $v \in V_i$ of a given length, and $\{I\} \times \ldots \times O(d_i) \times \ldots \times \{I\} \subseteq \mathcal{G}$, all vectors of length $|pr_i(u)|$ in $V_i$ must as well be in $U$. Since $U$ is closed under scalar multiplication and $pr_i(u) \neq 0$, this means that indeed $V_i \subseteq U$.   □

*Proof of Claim 2.* Each run of the given network is determined by sampled initial data:

- The initial weights $w_1, \ldots, w_M \in W = \mathbb{R}^d$ in the first layer,

- the inputs $x_1, \ldots, x_N \in V = \mathbb{R}^d$,

- the initial weights in the later layers, and biases in all layers, we will write them as one large vector $w' \in W' = \mathbb{R}^{d'}$,

- the targets $y_1, \ldots, y_N \in \mathcal{Y}$.

We will consider the targets as fixed and write the other initial data as

$$(w_i, x_j, w') \in \mathcal{V} := W^M \times V^N \times W'.$$

The group $\mathcal{G} \subseteq O(d)$ operates on $V = \mathbb{R}^d$ and $W = \mathbb{R}^d$, we also define its operation on $\mathcal{V}$ by

$$G : \mathcal{V} \to \mathcal{V}, (w_i, x_j, w') \mapsto (Gw_i, Gx_j, w') \tag{4}$$

By our assumptions and the definition of $\mathcal{G}$, this operation leaves the distribution of initial data invariant for any $G \in \mathcal{G}$. This proves the "$t = 0$" case in Claim 2, it remains to show that this invariance is kept when we do one step of Gradient Descent, this will follow if we can show that the update commutes with the operation of $\mathcal{G}$.

At each step of the training the current state is given by a point in the vector space $\mathcal{V}$, the loss is a function $L : \mathcal{V} \to \mathbb{R}$ and one step of Gradient Descent is given by

$$w_i \mapsto w_i - \epsilon \cdot \nabla_{w_i} L(w_i, x_j, w'), \qquad w' \mapsto w''$$

for some $w'' \in W'$. We have to show that this update commutes with the operation of $\mathcal{G}$, i.e. for all $G \in \mathcal{G}$

$$Gw_i \mapsto Gw_i - \epsilon \cdot G\nabla_{w_i} L(w_i, x_j, w'), \qquad w' \mapsto w'' \tag{5}$$

for the same $w''$.

Since the first layer only gets information about the $w_i$ and $x_j$ via their scalar products $\langle w_i, x_j \rangle$, and for any $G \in O(d)$ we have $\langle Gw_i, Gx_j \rangle = \langle w_i, x_j \rangle$, the operation (4) of $G \in \mathcal{G}$ on $\mathcal{V}$ leaves the loss function $L : \mathcal{V} \to \mathbb{R}$ invariant. For the same reason also the updates of the biases and later weights $w'$ are unaffected by the operation of $\mathcal{G}$. So we only have to prove the update equations for $Gw_i$, i.e. show that

$$\nabla_{w_i} L(Gw_i, Gx_j, w') = G\nabla_{w_i} L(w_i, x_j, w')$$

The gradients $\nabla_{w_i} L$ in the $w_i$–directions are part of the full gradient $\nabla L$ (which is a vector in $\mathcal{V}$, so it also contains the derivatives with respect to the points $x_j$ and the weights $w'$ of the later layers). Hence it is sufficient (or even stronger) if we can show

$$\nabla L(Gp) = G\nabla L(p) \qquad \text{for each point } p \in \mathcal{V} \text{ and } G \in \mathcal{G}. \tag{6}$$

Since $L$ is invariant under $G$, also the differential form $dL$ is invariant: $G^* dL = dL$.

The standard Euclidean metric on $\mathcal{V} = \mathbb{R}^{d \times M + d \times N + d''}$ provides a translation between differential forms and vector fields that determines $\nabla L$ from $dL$ and vice versa by the condition

$$\langle \nabla L(p), v \rangle = dL|_p(v) \qquad \text{for all } v \in \mathcal{V}.$$

Since the Euclidean metric on $\mathcal{V}$ is invariant under $G \in \mathcal{G}$, we have for each vector $v \in \mathcal{V}$

$$\langle \nabla L(Gp), Gv \rangle = dL|_{Gp}(Gv) = dL|_p(v) = \langle \nabla L(p), v \rangle = \langle G\nabla L(p), Gv \rangle$$

Since $G \in \mathcal{G}$ is invertible, $Gv$ can be any vector in $\mathcal{V}$, so (6) follows. $\qquad \square$

Note that the same proof argument holds for many other optimization techniques, not only for gradient descent. For example, it does not matter whether we use Momentum, AdaGrad, Adam, Nesterov, weight decay, etc. in the gradient descent. However, the optimization needs to respect symmetries from $O(d)$, i.e. it cannot make use of the special coordinate system for the input. This excludes for example Exponentiated Gradient [29].

Similarly, the proof is also independent of the loss function used, if that loss only involves the output of the network: Since the proof only makes use of symmetries $\langle Gx, Gw \rangle = \langle x, w \rangle$ in the first layer, it is not affected by what we do in the later layers or at the output level. However, if the loss function includes regularization terms that involve the weights in the first layer, these terms also need to be invariant under the orthogonal group. This means the proof still applies if we use $L2$ regularization, but $L1$ regularization does make use of the special coordinate system and is not invariant under rotations, so the proof arguments would not hold anymore if we used it in the first layer.

# D Measuring Alignment

## D.1 How well can we measure eigenvectors?

We do not observe the covariance matrices directly, but estimate them from samples of the corresponding distributions. This estimate has a variance that results in a variance of the computed eigenvectors. As a first intuitive experiment we compare the eigenvectors obtained from two disjoint samples:

**Data** We compare the covariance matrices $\Sigma_x$ and $\Sigma'_x$ obtained from $5 \times 5$ patches of the first and second $30\,000$ images in CIFAR10. The first eigenvectors are extremely well aligned:

| i | $\sigma_i^2$ | $|\langle e_i, e'_i \rangle|$ | i | $\sigma_i^2$ | $|\langle e_i, e'_i \rangle|$ |
|---|---|---|---|---|---|
| 1 | 12.3 | 0.9999996 | 7 | 0.28 | 0.99985 |
| 2 | 1.45 | 0.99995 | 8 | 0.23 | 0.99988 |
| 3 | 1.18 | 0.99992 | 9 | 0.128 | 0.99010 |
| 4 | 1.06 | 0.99994 | 10 | 0.126 | 0.99030 |
| 5 | 0.37 | 0.99969 | 11 | 0.115 | 0.99969 |
| 6 | 0.32 | 0.99970 | 12 | 0.096 | 0.99999 |

Also the other eigenvectors are very well aligned, all scalar products obtained were above 0.975. So 30,000 images are sufficient to estimate the eigenvectors of $\Sigma_x$ to a high accuracy.

**Weights** We use the same ResNet experiment as for Figure 3, described in Section A.7. We use two disjoint groups of 30 randomly initialized networks, trained in the same way on CIFAR10 with random labels.

| i | $\sigma_i^2$ | $|\langle e_i, e'_i \rangle|$ | i | $\sigma_i^2$ | $|\langle e_i, e'_i \rangle|$ |
|---|---|---|---|---|---|
| 1 | 0.281 | 0.974 | 7 | 0.085 | 0.906 |
| 2 | 0.202 | 0.958 | 8 | 0.078 | 0.946 |
| 3 | 0.154 | 0.916 | 9 | 0.077 | 0.924 |
| 4 | 0.141 | 0.934 | 10 | 0.072 | 0.811 |
| 5 | 0.107 | 0.861 | 11 | 0.063 | 0.598 |
| 6 | 0.104 | 0.861 | 12 | 0.059 | 0.645 |

So we can only measure the most important eigenvectors with reasonable accuracy. Of course it is not unexpected that we can determine the covariance of image patches much better than the covariance of weights: In the above example, we used $30000 \times 30 \times 30 = 27$ million image patches, but only $30 \times 64 = 1920$ weight vectors (filters). However, apart from the number of examples, also the type of information we want to extract from the covariance matrix determines the accuracy of our measurement.

In particular, when two eigenvalues are close together (like $\sigma_5, \sigma_6$ above), it may be difficult to determine the exact eigenvector. However, the 2-dim space spanned by both eigenvectors is relatively stable – in the above example the expansion of $e_5, e_6$ in terms of the basis $e'_i$ is

$$e_5 = 0.86 e'_5 - 0.36 e'_6 + ...(\text{smaller terms})...$$
$$e_6 = 0.36 e'_5 + 0.86 e'_6 + ...(\text{smaller terms})...$$

A similar observation can be made in figure 3: The first eigenvalues of $\Sigma_w$ are 0.019, 0.017, 0.011, 0.010, 0.009,..., with the first two close together; the corresponding two eigenvectors of $\Sigma_w$ given on the right of figure 3 were:

 and 

So the vector space of these two eigenvectors of $\Sigma_w$ is well aligned to the vector space of two eigenvectors of $\Sigma_x$, but its given basis is rotated compared to the eigenvectors of $\Sigma_x$.

This is a general problem - if two eigenvalues are close to each other, the direction of eigenvectors is not measurable in practice, or if they are even the same, also theoretically individual eigenvectors are not defined, only the higher dimensional eigenspace is. We will investigate this problem of statistical sampling uncertainty closer in D.3.

Apart from the statistical sampling uncertainty we may also consider noise in the images themselves. However, given that the two computations of the eigenvectors agreed so well for the data covariance, we can expect that this error is small compared to the sampling error for the weight covariance.

## D.2 Motivational definition of misalignment

Any attempt of defining "misalignment measures" naively by using the direction of eigenvectors (e.g. "misalignment = sum of angles between corresponding eigenvectors") runs into the problem mentioned above: It would be undefined when we have eigenspaces of dimension $> 1$ or at least discontinuous / not measurable / not informative in practice when two of the eigenvalues are close. So instead we will construct a "distance to an aligned matrix" that takes into account what can actually be measured.

In [23] a Riemannian metric is defined on the space of color perceptions. In this metric, the length of a path in color space is given by the minimal number of colors on this path connecting start and end such that each color is indistinguishable (by a human) from the next one.

We can use a similar construction for covariance matrices: Fix a (large) $n$, then we call two matrices indistinguishable, if after sampling $n$ samples from $\mathcal{N}(0, \Sigma_k)$ we most of the time cannot reject the hypothesis that the correct covariance matrix was $\Sigma_{k+1}$. The minimal number of "indistinguishable" matrices connecting start to end of a curve is (in good approximation) proportional to $\sqrt{n}$, to get a path length independent of $n$, we divide the number by $\sqrt{n}$ and take the limit $n \to \infty$. The misalignment could now be defined as the minimal path length of a path between $B$ and a matrix $\Sigma$ that is aligned to $A$. (The exact definition would need to specify "indistinguishable", i.e. the fraction of times we can reject the hypothesis and the confidence level used in this rejection. The path lengths corresponding to different definitions would differ by a multiplicative constant.)

The Riemannian metric defined in this way is given locally by the Fisher Information, which can be easily computed in this case. However, it seems there is not a simple known formula for the resulting (global) distance between two points. To simplify computations and proofs, we will use instead the upper bound to the square length given by the symmetrized Kullback–Leibler divergence. Like the square length of the shortest path between $A$ and $B$ it can be expressed as an integral over the Fisher Information, but the path is not the usual shortest path (geodesic) that is traversed with constant speed, but the straight line $(1-t) \cdot A + t \cdot B$. For small distances this is a good approximation, but it will give a larger value in general.

## D.3 Fisher information and Riemannian manifold of symmetric positive definite matrices

As a concrete toy example for the problem of close eigenvalues observed above, we take the 2-dimensional normal distribution with $\mu = 0$ and $\Sigma$ the diagonal matrix with entries $(1, \lambda)$. We generate $n = 100$ i.i.d. samples $\vec{w}_1, ..., \vec{w}_n$ of this distribution, measure their empirical covariance $\hat{\Sigma}_w$, and plot the direction of their eigenspaces. Repeating this 50 times, we can visualize the distribution of these directions (Figure 19, first row).

While we get a reasonable approximation to the direction of the eigenspaces of $\Sigma$ (i.e. the coordinate axes) for large $\lambda$, the eigenvector directions of $\hat{\Sigma}$ become less well-aligned as $\lambda = \sigma_2^2$ gets close to $1 = \sigma_1^2$. So if we were to measure "whether the diagonal matrix $diag(1, \lambda)$ is aligned with itself" by computing two empirical covariance matrices (from 2 independent sets of sampled data), their eigenvectors, and angles between the two empirical results, we would think that these two empirical covariances are not aligned for $\lambda \approx 1$, although they actually come from the same distribution.

Mathematically, the problem is that any "angles between eigenvectors" measure would be undefined for matrices with an eigenspace of dimension $> 1$. We could fix that e.g. by using the choice of eigenvectors that gives the smallest possible result, but then the resulting function would not be continuous around such matrices, i.e. we may need estimates with unlimited precision to get the alignment measure with a fixed precision.

On the other hand, for any fixed $\sigma_1^2 \neq \sigma_2^2$ we can in theory make $n$ large enough to get the sampling error as small as we want, e.g. in the above example we can go from $n = 100$ to $n = 100\,000$ (Figure 19).

Figure 19: Visualization of the variation of estimating the directions of the eigenspaces for 2-dimensional normal distribution with $\mu = 0$ and $\Sigma$ the diagonal matrix with entries $(1, \lambda)$ for three different values of $\lambda$ and varying number of samples $n$.

We can say that for $\lambda_1 \approx \lambda_2$ we have less information per sample from the normal distribution $\mathcal{N}(0, \Sigma)$ about the rotation angle $\alpha$ than for $\lambda_1 \gg \lambda_2$. This is formalized by the Fisher information, we now summarize some of its properties for the case we are interested in.

We denote by $M$ the manifold of symmetric positive definite $d \times d$ matrices, it is an open cone in the $D := \frac{d(d+1)}{2}$ dimensional vector space of symmetric matrices. We identify the points $\Sigma \in M$ also with their corresponding normal probability distributions $\mathcal{N}(0, \Sigma)$. Given $n$ samples $x_1, ..., x_n \in \mathbb{R}^d$, the empirical covariance matrix $\frac{1}{n} \sum_{i=1}^{n} x \cdot x^T \in M$ is the maximum likelihood estimator. Given some local coordinates $\theta_1, ..., \theta_D$ around a matrix $\Sigma$, the Fisher information of $\theta_k$ at $\Sigma$ is defined as

$$I(\theta_k) := \mathbb{E}_{x \sim \mathcal{N}(0, \Sigma)} \left[ \left( \frac{\partial \log p(x|\theta_k)}{\partial \theta_k} \right)^2 \right] \tag{7}$$

We are interested in the variance of the maximum likelihood estimate $\hat{\theta}_k$ when we estimate the parameter $\theta_k$ from $n$ samples $x_1, .., x_n$, it can be approximated as

$$Var(\hat{\theta}_k) \approx \frac{1}{n \cdot I(\theta_k)},$$

and this approximation becomes exact as $n \to \infty$ (see e.g. [50], chapter 9.7. for a precise statement). In our toy example we are interested in the rotation angle, and local coordinates would be $\alpha, \lambda_1, \lambda_2$ (we assume for the moment that $\lambda_1 \neq \lambda_2$ at $\Sigma$). In this case one can compute the Fisher information about $\alpha$ (see e.g. [12]) as

$$I(\alpha) = \frac{(\lambda_1 - \lambda_2)^2}{\lambda_1 \cdot \lambda_2}.$$

In the Figure 19, the images on the diagonal look similar, and indeed we have chosen $\lambda$ such that the Fisher Information about $\alpha$ of $n$ samples is approximately the same: In each case $n \cdot (\lambda - 1)^2 / (\lambda \cdot 1)$ is between 904.1 and 909.1, which gives a standard deviation of about 1/30 (corresponding to about 2 degrees) for the rotation angle $\alpha$, and indeed generating 100 directions from a normal distribution of $\alpha$ with a standard deviation of 1/30 matches the above plots on the diagonal.

This suggests that

$$\delta = \alpha \cdot \sqrt{I(\alpha)}$$

is an appropriate distance measure between matrices that differ by a rotation of angle $\alpha$: When two matrices $A, B$ have "distance" $\delta$, this means that for large $n$ we can put about $k = \lfloor \delta \cdot \sqrt{n} \rfloor$ points $\Sigma_1, \Sigma_2, ..., \Sigma_k$ between $A = \Sigma_0$ and $B = \Sigma_{k+1}$ such that each pair $\Sigma_i$ and $\Sigma_{i+1}$ are within measurement error of each other when our measurement consists of estimating the covariance from a sample of $n$ points.

So far, we have defined the Fisher information in a particular coordinate system on the manifold $M$. However, the definition (7) depends only on the tangent vector $\partial / \partial \theta_k$. So the Fisher information is actually assigned to tangent vectors of the manifold, is independent of the coordinate system, and it turns out to satisfy the properties of a Riemannian metric, so it gives for any tangent vector $v \in T_\Sigma(M)$ a "length" $\sqrt{I(v)}$. This is a special case of the general theory of Information Geometry, see e.g. [2, 5].

### D.4  Defining misalignment measures

While the Fisher information gives a local distance measure (i.e. a length of tangent vectors), we rather need a global distance measure $D(\Sigma_1, \Sigma_2)$ between points. For that we integrate the Fisher Information along the straight line

$$\gamma : t \mapsto (1 - t) \cdot \Sigma_1 + t \cdot \Sigma_2 \qquad \text{for } t \in [0, 1]. \tag{8}$$

If instead of (8) we had used the real shortest path (geodesic) of the Riemannian metric, which goes in constant speed from $t = 0$ to $t = 1$, this would give the square length of the shortest path. Since in general (8) is not the shortest path, and the parameterization does not have constant derivative with respect to the standard connection, this can give slightly larger results. Of course, for small distances it still gives the same result in first order.

In terms of information geometry, the straight line is the $e$–geodesic, and the integration of the Fisher information against the $e$–geodesic gives the same result as integrating along the $m$–geodesic: The symmetrized Kullback–Leibler divergence between $p$ and $q$ (see e.g. Theorem 3.2. of [2], also compare section 4.4.2. in [5]). The symmetrized Kullback–Leibler divergence exists in two normalizations: With or without the factor $1/2$. We are using the version with $1/2$

$$D(\Sigma_1, \Sigma_2) := \frac{D_{KL}\Big(\mathcal{N}(0, \Sigma_1) \,||\, \mathcal{N}(0, \Sigma_2)\Big) \; + \; D_{KL}\Big(\mathcal{N}(0, \Sigma_2) \,||\, \mathcal{N}(0, \Sigma_1)\Big)}{2}$$

which is equal to half of the integral of the Fisher information:

$$D(\Sigma_1, \Sigma_2) = \frac{1}{2} \int_0^1 I\left(\frac{\partial \gamma}{\partial t}(t)\right) dt$$

So we will use this distance measure as a basis for our (mis)alignment measure. It can be expressed analytically as

$$D(\Sigma_1, \Sigma_2) = \frac{\mathbf{tr}(\Sigma_1^{-1}\Sigma_2 + \Sigma_2^{-1}\Sigma_1)}{2} - d$$

While it would be possible to use different distance measures that also have the Fisher information as the infinitesimal version (e.g. the usual, asymmetric Kullback–Leibler divergence), this definition has the additional benefit that it is invariant under scaling: $D(\lambda\Sigma_x, \lambda\Sigma_w) = D(\Sigma_x, \Sigma_w)$, and this allows the simple definition of the alignment measure given in Section 2 (for more general distance measures one would need to restrict the $\Sigma$ to a subset e.g. by requiring $\mathbf{tr}(\Sigma) = 1$ and write $\inf_{\Sigma, \lambda > 0} D(\Sigma, \lambda \cdot \Sigma_w)$ to get a result $> 0$ when the matrices are not aligned).

So we define the "misalignment" score for two positive definite symmetric matrices $A, B$ as

$$M(A,B) := \inf_{\Sigma \succ 0 \text{ aligned with } A} D(\Sigma, B) = \inf_{\Sigma \succ 0 \text{ aligned with } A} \left\{ \frac{\mathbf{tr}(\Sigma^{-1}B + B^{-1}\Sigma)}{2} - d \right\},$$

which was our definition in Section 2.2.

**Proposition 2.** *This misalignment measure has the following properties:*
*For all positive definite symmetric matrices $A, B$*

1. *$M(A,B) \geq 0$*

2. *$M(A,B) = 0$ if and only if $B$ is aligned with $A$.*

3. *$M(A,B)$ is continuous in $B$.*

4. *Equivariance under orthogonal group: $M(UAU^T, UBU^T) = M(A,B)$ for $U \in O(d)$*

5. *Invariance under scalar multiples of $B$: $M(A, \lambda B) = M(A,B)$ for $\lambda > 0$*

6. *$M(A,B)$ only depends on the eigenspaces of $A$.*

7. *$M(A,B) + d = \sum_{i=1}^{r} \sqrt{\mathbf{tr}(B|_{V_i}) \cdot \mathbf{tr}(B^{-1}|_{V_i})}$, where $V_1 \oplus ... \oplus V_r$ is the orthogonal decomposition of $\mathbb{R}^d$ into eigenspaces of $A$, and $B|_{V_i}$ is the linear map $V_i \to V_i, \boldsymbol{v} \mapsto pr_i(B(\boldsymbol{v}))$ with $pr_i$ the orthogonal projection $\mathbb{R}^d \to V_i$.*

*The function $M(A,B)$ is not continuous in $A$, and there cannot be a function $M(A,B)$ that is continuous in both arguments and still satisfies condition 2.*

*Proof.* 1, 4, 5, 6, and the "if" part of 2 follow directly from the definitions.
3 follows from 7. We will see in the proof of 7 that the infimum is obtained for a particular matrix $\Sigma$, and from that also the "only if" part of 2 follows.
Formula 7: We use the orthogonal decomposition $V_1 \oplus ... \oplus V_r$ of $\mathbb{R}^d$ into eigenspaces of $A$. A positive definite symmetric matrix $\Sigma$ aligned with $A$ is given by its eigenvalues $\lambda_i$ on the subspaces $V_i$. For a linear map $f : \mathbb{R}^d \to \mathbb{R}^d$ we have

$$\mathbf{tr}(f) = \sum_{i=1}^{r} \mathbf{tr}(f|_{V_i})$$

So the definition of $M(A,B)$ can be rewritten

$$
\begin{aligned}
M(A,B) + d &= \frac{1}{2} \cdot \inf_{\lambda_1,...,\lambda_r > 0} \sum_{i=1}^{r} \mathbf{tr}\left(\lambda_i^{-1}B + \lambda_i B^{-1}\right)|_{V_i} \\
&= \frac{1}{2} \cdot \sum_{i=1}^{r} \inf_{\lambda > 0} \lambda^{-1}\mathbf{tr}(B|_{V_i}) + \lambda \mathbf{tr}(B^{-1}|_{V_i})
\end{aligned}
$$

Since $B$ is positive definite, $B^{-1}$ is positive definite as well. When $B$ is positive definite, then also $B|_V$ is positive definite, so we have $\mathbf{tr}(B|_{V_i}) > 0$ and $\mathbf{tr}(B^{-1}|_{V_i}) > 0$. Thus for $b := \mathbf{tr}(B|_{V_i})$ and $c := \mathbf{tr}(B^{-1}|_{V_i})$ the function $\lambda^{-1}b + \lambda c$ has a minimum for a finite positive $\lambda$, and it is

$$\arg\min_{\lambda > 0} \lambda^{-1}b + \lambda c = \sqrt{b/c}$$

as is seen e.g. by comparing the derivative with zero, and hence

$$\inf_{\lambda > 0} \lambda^{-1}b + \lambda c = \min_{\lambda > 0} \lambda^{-1}b + \lambda c = 2 \cdot \sqrt{b \cdot c},$$

which gives the above formula 7.

$M(A,B)$ cannot be continuous in $A$ if condition 2 should hold: Take $B$ the diagonal matrix with entries 1 and 2, and for $A$ consider the diagonal matrices with entries 1 and $\lambda$. Then for $\lambda = 1$ the matrix $B$ is not aligned with $A$, but it is for all $\lambda \neq 1$. Therefore we must have $M(diag(1,1), B) > 0$, but $M(diag(1,\lambda), B) = 0$ for all $\lambda \neq 0$, so $A$ would not be continuous with respect to $A$. $\qquad \square$

# E The shape of the transfer function $f(\sigma)$

## E.1 Gaussian centered input

In the "ideal" case of Gaussian centered input, we experimentally observe curves that "look continuous", see Figure 20. For this figure, we used different settings to obtain curves that go only down (left), rise and then fall (middle) or only go up (right).

We always used Gaussian input $\mathcal{N}(0, \Sigma_x)$ with mean 0 and a diagonal matrix for the covariance $\Sigma_x$ (which is no restriction of generality, since applying an orthogonal matrix to the input does not change the dynamic of the neural networks, and any symmetric positive definite matrix can be written as $O^T L O$ with diagonal matrix $L$ and orthogonal matrix $O$). The labels are always uniformly randomly sampled from $\{0, 1, ..., 9\}$. The networks are fully connected networks with 2 hidden layers and ReLU activation, we use cross entropy loss. We use standard He initialization and train with Gradient Descent until convergence. In each of the three plots of Figure 20 we show the results from 5 runs with different random initializations.

Settings for the three plots in Figure 20:

|  | Left | Middle | Right |
|---|---|---|---|
| Input dim | 10 | 30 | 30 |
| Layer 1 | 2048 | 2048 | 256 |
| Layer 2 | 256 | 256 | 256 |
| Output dim | 10 | 10 | 10 |
| Input size | 10 000 | 10 000 | 2 000 |
| Input $\Sigma_x =$ | $diag(1, 1.1, 1.2, ...1.9)$ | $diag(0.1, 0.2, ..., 3.0)$ | $diag(0.1, 0.2, ..., 3.0)$ |

We can give a heuristic argument for why the curves should "look continuous": If two eigenvalues of the data covariance $\Sigma_x$ are close, exchanging them gives an input distribution that is close to the original distribution. Because this exchange is an orthogonal transformation, we will get trained networks in which the weights also only differ by the same orthogonal transformation. This means that the effect of exchanging the sigmas will also exchange the taus, and if exchanging the sigmas had a small effect on the input distribution, we may expect also a small effect on the weights distribution, which means we expect the corresponding taus also to be close.

For the other experimental observation, that the curves first rise and then fall again (where one of these two parts can also be missing), we already sketched the two conjectured mechanisms in 2.3:

1. Larger eigenvalues $\sigma_i$ lead to larger effective learning rate in gradient descent, which leads in turn to larger corresponding $\tau_i$, hence the increasing part of $f$.

2. We find experimentally that the first eigenvector(s) dominate the input (see e.g. the first table in Appendix D.1). Using an orthonormal basis $e_i$ of eigenvectors of $\Sigma_x$ and $\Sigma_w$, we can decompose the variance of the output to a neuron $\mathbb{E}_x[\langle w, x \rangle^2]$ as $\sum_i \langle w, e_i \rangle^2 \mathbb{E}_x[\langle e_i, x \rangle^2] = \sum_i \langle w, e_i \rangle^2 \cdot \sigma_i^2$. Averaging over $w$ gives $\sum_i \tau_i^2 \cdot \sigma_i^2$. So if $f(\sigma)$ would be increasing, direction $e_1$ would dominate the output even more. We speculate that backprop finds a near optimal solution, and it seems plausible that one component dominating is not optimal when there is also important information in the other components.

Figure 20: $f(\sigma)$–curves for fully connected networks with two hidden layers and Gaussian centered inputs. See text for details.

Figure 21: $f(\sigma)$ curve for training on random labels with different approximations of CIFAR10 images, cropped to $27 \times 27$ pixels. Two $3 \times 3$-convolutional layers with stride 3, one fully connected layer. RIGHT: Zoomed in to the upper right corner to show the small differences.

Figure 22: $f(\sigma)$ curve for training on random labels with different approximations of CIFAR10 images. Two $3 \times 3$-convolutional layers, one fully connected layer. Correlations between neighboring patches create deviation from the curve seen for the "ideal" case of independent Gaussian approximations. Smaller strides lead to stronger correlations and stronger deviation.

## E.2    Convolutional networks on natural images

While the $f(\sigma)$ curves in the ideal centered Gaussian case look smooth, the real-world $f(\sigma)$ curves seem to contain perturbations, although they still roughly have a rising/falling shape as well. To investigate what the largest contributor to this perturbations is, we will go from the ideal case to the real case in a series of steps.

As a first step, we crop the images of CIFAR10 to $27 \times 27$ pixels and approximate their distribution by the normal distribution with mean 0 and the covariance of all $3 \times 3$ patches. When we apply a convolutional network with $3 \times 3$ convolutions and stride 3, we are in the "ideal" situation of E.1 and expect a $f(\sigma)$ curve that goes up and then down. This is indeed the case, as shown by the green curve in Figure 21. Using the same distribution / covariance for all patch positions is of course a simplification: For example, for the patches at the upper boundary it is more likely to see light blue (from the sky) than at the lower boundary. To get closer to reality, we can replace the one global covariance by the covariances corresponding to the possible patch positions, yielding the red curve in Figure 21.

The next step is to abandon Gaussian approximation, and take the real patches. In the $27 \times 27$ pixel images we have $9 \times 9$ patches of $3 \times 3$ pixels each. To destroy the correlations between neighboring patches, we permute for each of the $9 \times 9$ positions the patches in that position of all images. So for each position we still have the same set of $3 \times 3$ patches that can appear in an image, but the patches appearing in one new image no longer fit together since they (almost always) came from different original images. This leads to images that are stitched together from random patches and also means that previously (potentially similar) patches from the same original image can now occur with two different random labels. This setting yields the black curve in Figure 21. So far it seems we are still very close to the ideal situation.

The next step is to take the original images; now the correlations between neighboring patches do create a more significant change in the $f(\sigma)$ curve, see the blue curve in Figure 22. In particular, we see the first significant deviation from "up and then down". This effect becomes stronger when we go from stride 3 to overlapping inputs of the convolution (stride 2 and stride 1, green and red curves in Figure 22).

Figure 23: Sequence of $\tau_1, ..., \tau_{16}$ used to weight the 16 most significant eigenvectors in our experiments. The sequence $f_1$ was used for Table 1 in section 2.5, $f_2$ and $f_3$ were used for Table 2 and Table 3.

### E.3 Deeper Layers

We use the following simple CNN on the CIFAR10 data set here (as in Section 2.5, Table 1):

```
conv 3x3, 16 filters
conv 3x3, 16 filters
conv 3x3, 16 filters
maxpool 3x3, stride 2
dense layer, 512 units
dense layer, num_outputs units (classifier head)
```

Training is done using 30k images for pre-training on random labels (for comparison run), 20k images for training on real labels, 10k images for determining the test error, and using a learning rate of 0.002.

**Sampling from covariance vs. picking eigenvectors** In Figure 5 we compared a pre-trained convolutional layer with a layer consisting of filters randomly sampled from the same covariance matrix and found that this preserved the performance benefit exactly. In the experiment for Table 1 we instead used the most significant eigenvectors $e_1, ..., e_{16}$ with factors $\tau_1, ..., \tau_{16}$ directly as filters. This is a (more stable) approximation to sampling from the covariance matrix that has $\tau_i^2$ as eigenvalue for $e_1, ..., e_{16}$ and $\tau = 0$ for the other eigenvectors. Experimentally, we can see that this does not seem to make a significant difference: Compare Table 2 with Table 4 for the case that $\tau_1 = ... = \tau_{16} = 1$.

**Choice of eigenvectors $e_1, ..., e_{16}$ and $\tau_1, ..., \tau_{16}$** In our previous experiments we observed curves for $f(\sigma)$ that were "rising and then falling", favoring the most significant eigenvalues, but somewhat downweighting the most significant one(s) (e.g. the center and right plot in Figure 4 – this was a different network on the same data). So we picked the 16 largest $\sigma$ to be the ones with $f(\sigma) > 0$ (i.e. used the most significant eigenvectors), and used a made-up curve ($f_1$ in Figure 23) for the $\tau_1, ..., \tau_{16}$ which downweights the largest eigenvalue. We can see that the choice of this curve does not make a big difference: We can also just set $\tau_1 = ... = \tau_{16} = 1$ ($f_2$, resulting in Table 2), or choose another curve which downweights the largest eigenvalue less ($f_3$, resulting in Table 3). The stronger downweighting of the most significant eigenvalue seems to give a slight advantage when only initializing the first layer, but when initializing two or three layers, all curves seem to lead to essentially the same performance. However, compared to adjusting some values $f(\sigma)$ between 1 and 4, the much more important choice is which $f(\sigma)$ we set to 0, i.e. which eigenvectors we use. Tables 5 and 6 show the effect of choosing other eigenvectors, which leads to a significant drop in performance (for the choices in these tables even below the standard initialization).

Table 2: Using $f_2 = 1$: The 16 eigenvectors with largest eigenvalues are equally weighted. This corresponds to a $f(\sigma)$ which is 1 for the 16 largest $\sigma$s and 0 for the other (11 or 128) $\sigma$s. We achieve significant gains compared to the standard initialization.

| Iterations | Data | \{\} | \{1\} | \{1,2\} | \{1,2,3\} |
|---|---|---|---|---|---|
| | | Convolutional layers sampled | | | |
| 100 | Train | 0.31 | 0.29 | 0.37 | 0.44 |
| | Test | 0.31 | 0.29 | 0.37 | 0.43 |
| 1000 | Train | 0.58 | 0.59 | 0.65 | 0.68 |
| | Test | 0.53 | 0.55 | 0.56 | 0.59 |

Table 3: Using $f_3$: Changing the $f(\sigma)$ between 1 and 4 does not affect the performance significantly, as long as we keep the same set of $\sigma$ with $f(\sigma) > 0$.

| Iterations | Data | \{\} | \{1\} | \{1,2\} | \{1,2,3\} |
|---|---|---|---|---|---|
| | | Convolutional layers sampled | | | |
| 100 | Train | 0.31 | 0.30 | 0.37 | 0.44 |
| | Test | 0.31 | 0.29 | 0.37 | 0.42 |
| 1000 | Train | 0.58 | 0.61 | 0.64 | 0.72 |
| | Test | 0.53 | 0.55 | 0.58 | 0.56 |

Table 4: Sampling from a covariance matrix using $f(\sigma) = 1$ for the highest 16 eigenvalues and $f(\sigma) = 0$ else. This gives essentially the same performance as using the 16 most significant eigenvectors directly as filters.

| Iterations | Data | \{\} | \{1\} | \{1,2\} | \{1,2,3\} |
|---|---|---|---|---|---|
| | | Convolutional layers sampled | | | |
| 100 | Train | 0.31 | 0.29 | 0.37 | 0.41 |
| | Test | 0.31 | 0.29 | 0.36 | 0.40 |
| 1000 | Train | 0.58 | 0.57 | 0.62 | 0.69 |
| | Test | 0.53 | 0.54 | 0.54 | 0.56 |

Table 5: Using the 16 eigenvectors with the smallest eigenvalues gives significantly worse performance than random initialization. The 16 chosen eigenvectors were given the same weight.

| Iterations | Data | \{\} | \{1\} | \{1,2\} | \{1,2,3\} |
|---|---|---|---|---|---|
| | | Convolutional layers sampled | | | |
| 100 | Train | 0.31 | 0.15 | 0.14 | 0.15 |
| | Test | 0.31 | 0.15 | 0.14 | 0.15 |
| 1000 | Train | 0.58 | 0.50 | 0.48 | 0.47 |
| | Test | 0.53 | 0.46 | 0.47 | 0.44 |

Table 6: Using eigenvectors $e_4, ...e_{19}$ (out of 27) in the first layer, $e_{10}, ..., e_{25}$ (out of 144) in the second and third layer. The 16 chosen eigenvectors were given the same weight.

| Iterations | Data | \{\} | \{1\} | \{1,2\} | \{1,2,3\} |
|---|---|---|---|---|---|
| | | Convolutional layers sampled | | | |
| 100 | Train | 0.31 | 0.17 | 0.20 | 0.27 |
| | Test | 0.31 | 0.18 | 0.20 | 0.26 |
| 1000 | Train | 0.58 | 0.56 | 0.60 | 0.62 |
| | Test | 0.53 | 0.53 | 0.56 | 0.59 |

Table 7: Base line: These are the accuracies we obtain by pre-training on random labels. In particular when we use two or three convolutional layers, the results are very similar to what we get with our method that only uses the data, no training. (Compare e.g. with Table 2)

| Iterations | Data | \{\} | \{1\} | \{1,2\} | \{1,2,3\} |
|---|---|---|---|---|---|
| | | Convolutional layers sampled | | | |
| 100 | Train | 0.31 | 0.37 | 0.39 | 0.41 |
| | Test | 0.31 | 0.34 | 0.37 | 0.39 |
| 1000 | Train | 0.58 | 0.70 | 0.75 | 0.78 |
| | Test | 0.53 | 0.51 | 0.53 | 0.57 |

# F    Specialization at the later layers

Earlier, we claimed that neural activations at the outer layer can drop abruptly and permanently after switching to the downstream task. Figure 24 illustrates this. In this figure, the $x$-axis corresponds to the number of training iterations, where the transfer to the downstream tasks happens at the middle. Note that in both network architectures shown in the figure, neural activation in the upper layers drops abruptly and permanently after switching to the downstream tasks. We interpret this to effectively reduce the available capacity for the downstream task, which masks the positive transfer happening from the alignment effect at the lower layers. This effect can be mitigated by increasing the width, as discussed in the context of Figure 8 in the main text.

Regarding specialization, Figures 10 and 11 are the extended versions of Figure 8. They include activation plots for *all intermediate layers* of VGG16 models (i) at initialization, (ii) in the end of the pre-training, (iii) in the end of fine-tuning on real labels, (iv) in the end of fine-tuning on 10 random labels. Figures 10 and 11 illustrate the negative and positive transfer examples respectively. See Section A.11 for details about the parameters used when generating those figures. As shown in those figures, neurons at the upper layers tend to *specialize*, i.e. become activated by fewer images. This is evident if we compare the fraction of examples activating neurons when pretrained for 1 training iteration (TOP-LEFT) vs. pre-trained for 6240 training iterations (TOP-RIGHT).

Because neural activations drop permanently after switching to the downstream task, the capacity of the neural network in the downstream task is effectively diminished. This is reminiscent of the "critical stages" that have been observed for deep neural networks, in which neural networks seem to possess less capacity to fit a new distribution of data (e.g. after image blur is removed) later during training than when trained from scratch. Here, the upstream task is analogous to the critical early stage and the switch to the downstream task is analogous to the change of distribution (e.g. removing blur in images).

Figure 24: TOP: Neural activation is plotted against the number of training iterations in a two-layer CNN (256 $3 \times 3$ filters followed by a dense layer of width 64). The $y$-axis is the frequency of the output of a hidden activation function being non-zero measured over a hold-out dataset. The abrupt drop in neural activation coincides with the switch to the downstream task. BOTTOM: A similar plot for a deeper neural network with three convolutional layers (256 $3 \times 3$ filters) followed by a single dense layer of width 64.