[Reviews · NeurIPS 2020]

Review 1

Summary and Contributions: Training deep neural networks with entirely random labels have some interesting phenomenons. Previous work usually study this phenomenon from the perspective of memorization and generalization of neural network. In contrast, this paper studies an interesting phenomenon that *sometimes* pre-trained on randomly labels dataset can still help the transfer learning ability. This paper show the networks can learn weights that have principal components aligned with those from data, which may be the explanation of such positive effect. By providing theoretical analysis and empirically study, they verify such hypothesis. Furthermore, this paper also gives a possible explanation of cases where pretraining on random labels is harmful.

Strengths: - This paper is well-motivated, from an interesting phenomenon of training network with random label. - The story and mathematical derivations are easy to follow.

Weaknesses: I’m mainly concerned about the experiment session. In particular, I’m a bit confused about the experiment setting of Section 2.5. What is the number of layers in this setting when trained with random labels? From my perspective, when applying to downstream tasks, only the first layer’s weight is learned, and the deeper layers’ weights are analytically computed. Is that the case? ========= after rebuttal ========= I read the authors' rebuttal and decide to maintain my original score.

Correctness: Yes

Clarity: Yes

Relation to Prior Work: Yes

Reproducibility: Yes

Additional Feedback:


Review 2

Summary and Contributions: ==========Update after rebuttal========== I have read the author's response and the other reviews. The rebuttal addressed most of my concerns, although I still do not fully understand the link to critical periods. I look forward to the expanded discussion of this relationship in the camera-ready version. I leave my score unchanged and maintain that the paper should be accepted. =================================== This work is motivated by the observation that pretraining a deep network on randomly labelled images can either have positive or negative transfer effect on a downstream task, depending on the scale of the initialization and the number of downstream classes. They explain this observation by analyzing what deep networks learn when trained on random labels, finding that the first layer will learn to align the covariance of the weights to the input covariance.

Strengths: * I think this paper is highly significant and will be of mass appeal to the NeurIPS community. * The ideas are clearly communicated. Figure 3 is especially great. * The experimental details are throughly described in the supplemental material.

Weaknesses: * The formal analysis focuses only on the first layer. (This is not a major weakness.) In section 2.5, experiments with deeper layers are discussed. This discussion assumes that what's going on at the first layer is representative of deeper layers as well. However, the paper also discusses the task specification that occurs at deeper levels. It seems there will be a trade off between the bottom-up eigenspace alignment and the top-down task-specificity. The theoretical relationship between these two forces is left for future work. * The authors propose no broader societal or ethical impacts of their work. I think an argument can be made that work on understanding deep learning can contribute to interpretable and explainable AI, which has potential to help identify sources of bias, for example. I encourage the authors to try to write something more meaningful in that section.

Correctness: * It is claimed that the present results help to explain critical learning stages in DNNs. This claim is made in both the introduction and the discussion but it is not justified clearly anywhere in the text.

Clarity: The paper is well written and I did not find any typos or ungrammatical sentences. Minor points * Figure 4: use subtitles for each subfigure to make more readable e.g. synthetic, CIFAR10-random, CIFAR10 real. The labels are also too small to read.

Relation to Prior Work: * The authors fail to mention previous work on unsupervised pretraining. Back in the Deep Belief Network days, we thought unsupervised pretraining was necessary for deep learning to work. How do these results on what neural networks learn when trained on random labels mesh with our understanding of how unsupervised pretraining helped DBN training? * Erhan, D., Courville, A., & Vincent, P. (2010). Why Does Unsupervised Pre-training Help Deep Learning? Journal of Machine Learning Research, 11, 625–660. * Section 2.5 sounds similar to the procedure described in Dehmamy, N., Rohani, N., & Katsaggelos, A. K. (2019). DIRECT ESTIMATION OF WEIGHTS AND EFFICIENT TRAINING OF DEEP NEURAL NETWORKS WITHOUT SGD. In Proceedings of the IEEE International Conference on Audio, Speech and Signal Processing (ICASSP) (pp. 3232–3236). * The submission described a large body of work aimed at "identifying fundamental differences between real and random labels". This seems like a strange way of summarizing the goals of those works, which are more about identifying fundamental differences between networks that generalize and networks that memorize. But I understand the intended distinction between that work and the present submission, namely that previous work treats memorization as a negative thing whereas this submission also looks at positive transfer.

Reproducibility: Yes

Additional Feedback:


Review 3

Summary and Contributions: This paper aims to understand what a neural network learns when it is trained on random labels. As a motivating factor, the authors show that pre-training on random labels can speed up training on true labels. The explain this speed up with 'alignment' between the input data and the first layer weights.

Strengths: 1. The paper studies an interesting question - understanding what the network learns even when it is trained on random labels can give us more insight into the implicit biases of our architectures and training procedures - a largely open question. 2. Though Proposition 1 has been proven for a narrow case (Gaussian inputs), the conclusion seems to hold broadly as demonstrated by the experiments on sampling the weights from the Gaussian approximation. I found it surprising that you can gain speed ups only from the second order statistics of the weights. The empirical definition for misalignment also seems well thought out. 3. The discussion on specialization in the neurons in the later layers is also interesting and may have implications for transfer learning in general

Weaknesses: The main weakness of this paper is the organization and clarity. More detailed comments below. The paper also makes some unsubstantiated claims in some places. For instance, the paper provides some reasoning for the shape of f(\sigma) which could be tested but it is not easy to follow - why is it reasonable to expect that the large eigenvalues will dominate the output of the layers or what does it mean for backprop to capture this signal? ___ Post rebuttal ___ I am happy with the author response and would keep my score as is

Correctness: 1. Proposition 1 and the accompanying proof are correct 2. Experimental methodology is also correct

Clarity: The paper conveys the main points but can be organized better. The paper conducts a few different types of experiments and the results feel a bit scattered at times. Some more concrete problems: 1. The existence of positive and negative transfer has been mentioned a few times, but the precise cases in which they arise have not been specified. 2. f(\sigma) is not properly defined. The experimental setup for the synthetic case is not properly described. 3. Line 213 "Suppose that instead of pre-training on random labels, we sample from the Gaussian approximation of the filters in the first layer that were trained on random labels." It is hard to understand what the exact procedure is here.

Relation to Prior Work: Yes

Reproducibility: Yes

Additional Feedback:


Review 4

Summary and Contributions: This paper analyzes the effect of training with random (fixed) labels on the weights of a neural network. The authors show theoretically and empirically that the principal components of the weights align to those of the data. The experiments are carried out on three different architectures using the CIFAR-10 dataset.

Strengths: Understanding the effects of training with random labels is an important problem that is worthwhile investigating. This paper makes an interesting observation both theoretically and practically. The theoretical findings are evaluated thoroughly using different architectures. The paper is well written and good to follow.

Weaknesses: Even though the authors dismiss performing experiments with image augmentations (L 50) as it would introduce a supervisory signal, it could be beneficial to investigate it in the paper. Even though augmentations do add a prior on the expected data distribution, it could be worthwhile to investigate the effect. This is of course another step away from the i.i.d. assumption in Proposition 1, but since neighboring patches of the same image are already correlated, the effect in practice could be interesting to observe. Along the same lines, I would expect that with increasing kernel size of the convolutions, the correlation between patches increases and with that potentially the misalignment score. If this understanding is correct I would also expect that the experiment in Fig. 6 would look very different if only the last layers were transferred instead of the first layers. This would mean that random labels are a meaningful proxy to learn early layers but not the later layers. This would also align with the specialization observation in Sec. 3. +++++++ Post Rebuttal +++++++ After reading the other reviews and the authors’ feedback, I find most concerns addressed and will keep my recommendation. However, I would still recommend investigating the effect of augmentations on the training, since these are crucial in almost all applications and will affect the independence assumption between patches.

Correctness: The claims of the paper are theoretically sound and are empirically validated in several experiments. Due to the randomness involved in NN training (weights, batches) results are reported as averages over multiple runs in most places.

Clarity: The paper is well written and the structure makes sense, unraveling different observations and experiments, one at a time. Section 3, however is not fully self contained and has moved a lot of content to Appendix F. Similarly, the paper contains only CIFAR-10 evaluations, all ImageNet experiments are in the appendix.

Relation to Prior Work: The related work section is well structured and contains most relevant literature. One angle that could be included is the area of research that investigates what early layers learn by reducing the amount of training data that is used (eg 1 image in [Asao20], handcrafted Conv 1+2 via scattering transforms in [Oyallon18]). This direction is interesting as it shows that the early layers of a CNN do not seem to depend much on the actual training data and can be learned from little data or directly crafted. The paper here comes to a similar conclusion from the theoretical side. References A critical analysis of self-supervision, or what we can learn from a single image Yuki M. Asano, Christian Rupprecht, Andrea Vedaldi ICLR 2020 Scattering networks for hybrid representation learning. Oyallon, Edouard, Sergey Zagoruyko, Gabriel Huang, Nikos Komodakis, Simon Lacoste-Julien, Matthew Blaschko, and Eugene Belilovsky TPAMI 2018

Reproducibility: Yes

Additional Feedback: With the findings in the paper it should be possible to construct a set of weights that aligns with the observed data distribution. Would this constructed set of weights be enough to accelerate downstream training? Typos and other minor things: Fig.4 “per epochs” -> “per epoch” Text in Figs. 3&4 is very small

[Author Response · NeurIPS 2020]

We would like to thank the reviewers for their careful and positive reviews. Due to space constraints, we respond to
some of the reviewers' comments and questions here. All comments will be addressed in the final version of the paper.

Reviewers 2 and 4 suggest discussing additional related work, such as unsupervised pretraining and reducing the amount
of pre-training data. We agree that these are relevant and will include them in the discussion of related work.

**Reviewer 1**   **Q:** "*I'm a bit confused about the experiment setting of Section 2.5 [...]*" **A:** In the setting of Tables 1-7,
we had 3 conv layers, and one fully connected layer, but in general we can look at the first $L$ conv layers in a neural
network with $\geq L$ conv layers. All of the layers are computed from the input images (without using labels or learning).
The simplest procedure (which already gives the main benefit, cf. Table 2) is this: The first layer uses the training input
directly, computes the covariance matrix $\Sigma_x$ of patches, and uses as $k$ conv filters the eigenvectors $e_i$ (normalized to
length 1) corresponding to the $k$ largest eigenvalues $\sigma_i^2$ of $\Sigma_x$. After computing this first layer, we can also compute the
output of the first layer on the training images, which then becomes the input to the second layer. Then we compute the
second layer in the same way using these transformed representations as input, and so on. Variants of this procedure
include multiplying the filters $e_i$ with $\tau_i = f(\sigma_i)$ for some assumed function $f(\sigma)$, or sampling the filters from a normal
distribution $\mathcal{N}(0, \Sigma_w)$ where $\Sigma_w$ has eigenvectors $e_i$ with eigenvalues $\tau_i^2$. These variants give very similar results (see
Tables 1-4).

**Reviewer 2**   **Q:** "*It seems there will be a trade off between the bottom-up eigenspace alignment and the top-down
task-specificity. The theoretical relationship between these two forces is left for future work.*" **A:** Thank you for this
accurate summary. We agree with your assessment.

**Q:** "*The authors propose no broader societal or ethical impacts of their work. I think an argument can be made that
work on understanding deep learning can contribute to interpretable and explainable AI, which has potential to help
identify sources of bias, for example. I encourage the authors to try to write something more meaningful in that section.*"
**A:** We will extend the discussion in the *Broader Impact* section following your advice.

**Q:** "*It is claimed that the present results help to explain critical learning stages in DNNs. This claim [...] is not justified
clearly anywhere in the text.*" **A:** Our experiments suggest that critical learning periods that reduce learning capacity (as
described in detail in Achille et al. [1]) occur partially because of the specialization that takes place at the upper layer.
In Figures 7, 10 & 11, we show that neurons at the upper layer do specialize; meaning that a single neuron tends to be
activated by a few images in the training sample only. This has a negative effect when the distribution of data changes.
Figure 24 in Appendix F shows that neural activation at the upper layer can drop abruptly and permanently once the
switch to the downstream task takes place. We will add an additional brief explanation in the paper.

**Reviewer 3**   **Q:** "*The paper provides some reasoning for the shape of $f(\sigma)$ which could be tested but it is not easy to
follow—why is it reasonable to expect that the large eigenvalues will dominate the output of the layers or what does it
mean for backprop to capture this signal?*" **A:** We find experimentally that e.g. for $5 \times 5$ patches on CIFAR10 $\sigma_1^2 \approx 12$
and $\sum_{i=2}^{75} \sigma_i^2 \approx 6$, so the first direction dominates the input (see first table in Appendix D.1). Using an orthonormal
basis $e_i$ of eigenvectors of $\Sigma_x$ and $\Sigma_w$, we can decompose the variance of the output to a neuron $\mathbb{E}_x[\langle w, x \rangle^2]$ as
$\sum_i \langle w, e_i \rangle^2 \mathbb{E}_x[\langle e_i, x \rangle^2] = \sum_i \langle w, e_i \rangle^2 \cdot \sigma_i^2$. Averaging over $w$ gives $\sum_i \tau_i^2 \cdot \sigma_i^2$. So if $f(\sigma)$ would be increasing,
direction $e_1$ would also dominate the output. We speculate that backprop finds a near optimal solution, and it seems
likely that one component dominating is not optimal when there is also important information in the other components.

**Q:** "*The existence of positive and negative transfer has been mentioned a few times, but the precise cases in which they
arise have not been specified.*" **A:** As Reviewer 2 mentioned, there is a tradeoff between both effects: the positive
effect at the early layers and the negative effect at the later layers. Depending on the setting, such as initialization (cf.
Figure 1) or the DNN architecture (cf. Figure 8), one effect may dominate the other.

**Q:** "*$f(\sigma)$ is not properly defined. The experimental setup for the synthetic case is not properly described.*" **A:** We will
make the definition of $f(\sigma)$ (Equation (2)) and the synthetic experimental setup in Appendix E.1. more explicit.

**Reviewer 4**   **Q:** "*I would expect that with increasing kernel size of the convolutions, the correlation between patches
increases and with that potentially the misalignment score. [...] This would also align with the specialization observation
in Sec. 3.*" **A:** Yes, we believe that correlation effects can be observed. Some results in this direction are presented in
Figure 22 in Appendix E.2, where we observe that "*Correlations between neighboring patches create deviation from
the curve seen for the "ideal" case [...]. Smaller strides lead to stronger correlations and stronger deviation.*"

**Q:** "*With the findings in the paper it should be possible to construct a set of weights that aligns with the observed
data distribution. [...]*" **A:** Yes, exactly. This is done in an example in Section 2.5. and we are considering this as an
interesting future direction of research.

[Meta-Review · NeurIPS 2020]

Reviewers unanimously agree that the paper presents insights that explain when training with random labels can help in learning transferrable features. The findings of this paper will be of broad interest to the ML community. I recommend that authors revise their paper in accordance to their comments in the rebuttal.